



# Adjoint-based calibration of inlet boundary condition for atmospheric computational fluid dynamics solvers

**Siamak Akbarzadeh**[1], **Hassan Kassem**[2], **Renko Buhr**[1], **Gerald Steinfeld**[1], and **Bernhard Stoevesandt**[2]

[1]ForWind, Center for Wind Energy Research, Carl von Ossietzky University Oldenburg,
Küpkersweg 70, 26129 Oldenburg, Germany
[2]Fraunhofer Institute for Wind Energy Systems – Fraunhofer IWES,
Küpkersweg 70, 26129 Oldenburg, Germany

**Correspondence:** Siamak Akbarzadeh (siamak.akbarzadeh@uni-oldenburg.de)

**Abstract.** A continuous adjoint solver is developed for calibration of the inlet velocity profile boundary condition (BC) for computational fluid dynamics (CFD) simulations of the neutral atmospheric boundary layer (ABL). The adjoint solver uses interior domain wind speed observations to compute the gradient of a calibration function with respect to inlet velocity speed and wind direction. The solver has been implemented in the open-source CFD package OpenFOAM coupled with the local gradient-based "CONMIN-frcg" solver of the DAKOTA optimization package. The feasibility of the optimizer output is continuously monitored during the calibration process. The inlet flow profile is considered acceptable only if it can be fitted to a logarithmic or power law function with a tolerance of 3 %. Otherwise, the optimization takes the last fitted profile and asks for a new gradient evaluation. The newly developed framework has been applied in two cases, namely the Ishihara case and Kassel domain. By using the measurements over the hill in the Ishihara case, the method was able to predict the velocity profiles upstream and downstream of the hill accurately. For the Kassel domain, despite the complexity of the site, the method managed to achieve the targeted profile within a reasonable number of the solver calls.

## 1 Introduction

The wind energy industry is growing very fast and a comprehensive site assessment is a key factor in planning, installation and performance of wind farms. As a result, the interaction of the atmospheric boundary layer (ABL) flow and wind turbines is one of the most important aspects of the site assessments for wind farms. Over the last decades with the development of powerful computers the computational fluid dynamics (CFD) has become one of the leading tools for microscale simulation of the wind flow over complex terrains. However, in general, developing algorithms that capture all the physics of such complex flow regimes is an ongoing research in CFD.

In CFD simulations of atmospheric flows the correct boundary conditions (BCs) are often unknown but measurement data (wind speed, wind direction, etc.) within the area of interest is available. Using the measurement data and an inverse analysis, the unknown BCs can be obtained with an optimization algorithm. This approach is called open boundary optimization and has been successfully tried in oceanography (Seiler, 1993; Chen et al., 2013) and numerical wind prediction (NWP) models (Schneiderbauer and Pirker, 2011). Another approach is to use observations and statistical analysis (Glover et al., 2011) to calibrate the CFD model parameters (e.g., inflow and turbulence model constants).

The solution to an optimization problem can be found with different methods. However, the methods such as genetic algorithm and evolutionary strategies (Davis, 1991; Michalewicz, 1996) require a large number of function evaluations which in CFD applications can be computationally very expensive. Alternatively, the gradient-based optimizers (Ruder, 2016) use the derivative of cost function with respect to the design parameters. Then, the optimal solution can be obtained using the gradient and a relatively less cost

function evaluation. Gradient computation with the finite-difference (FD) method is relatively simple. However, for a high number of design parameters it is prohibitively expensive. In the adjoint method, the sensitivity of the objective function can be calculated independently from the number of parameters, and this considerably reduces the cost of computation.

Since the first application of the adjoint method into compressible CFD models by Pironneau (1974), the adjoint-based optimization methods have been extensively used in shape and topology optimization (Jameson et al., 1998; Kämmerer et al., 2003; Li et al., 2006; Othmer and Grahs, 2005; Othmer et al., 2006). The adjoint gradient computation can be categorized into two main groups: (1) continuous adjoint and (2) discrete adjoint. In the continuous method (Jameson, 1988), the adjoint equations are first derived and then discretized. In the discrete adjoint (Giles and Pierce, 2000), using the chain rule, the adjoint solver is derived by line-by-line differentiation of the discretized original (primal) flow solver code. The discrete adjoint differentiation can also be automated using algorithmic differentiation (AD) (Griewank and Walther, 2008).

In principle, both methods can be applied to any algorithm and model which is continuously differentiable. However, the manually derived discrete adjoint differentiation of big CFD codes is laborious and error prone. Although, the AD tools can be seen as an interesting solution to this problem, their application to the codes which are written in high-level languages (e.g., C++) is still limited in terms of memory requirement and performance. A comparison of these two adjoint approaches can be found here (Giles and Pierce, 2000; Nadarajah and Jameson, 2000, 2001).

The discrete adjoint version of OpenFOAM-based solvers has been presented before (Towara and Naumann, 2013; Towara et al., 2015). More recently a hybrid approach has also been introduced (He et al., 2018) in which some parts of the code are differentiated by finite difference, and better performance is reported in comparison to the pure discrete adjoint version of the code. Despite all these improvements, the continuous adjoint version of OpenFOAM solvers are still more popular.

The adjoint method has been well applied to different problems in atmospheric science such as wind turbine blade shape optimization (Dhert et al., 2017), wind-farm control (Goit et al., 2016; Munters and Meyers, 2017; Vali et al., 2019) and wind-farm layout optimization (King et al., 2017). Bauweraerts and Meyers (2018) have used the adjoint method and large eddy simulation (LES)-based 4D-Var data assimilation to estimate the turbulent flow field of an atmospheric boundary layer domain from lidar data. Although compared to the Reynolds Averaged Navier–Stokes (RANS) models, the LES can provide more accurate predictions of the flow field, it is still mainly used as a research tool due to its demanding computing power.

The effect of inflow boundary wind speed and its direction are significant parameters in ABL CFD simulation, but they are not often known. The focus of this paper is on the adjoint gradient-based calibration of the inlet velocity profile and inflow wind direction (WD) for a RANS-based ABL flow, which is a very common CFD solver in the wind energy industry, with only a few wind speed measurements from a metrological mast at the site. The available continuous adjoint solver of OpenFOAM CFD tool package (*adjointShapeOptimizationFoam*), which is for topology optimization of duct flows, is further developed to compute the gradients.

The structure of the paper is as follows: the gradient-based optimization and the theory of the adjoint method are briefly presented in Sect. 2. The derivation of the adjoint equations and its BCs from the primal flow model is explained in Sect. 3. Finally, the numerical results and the conclusions are presented in Sects. 4 and 5.

## 2   Gradient-based optimization and adjoint method theory

In gradient-based optimization or calibration, one needs to compute the gradient of a smooth cost function, $J$, with respect to the design parameters, $\alpha$, at each design step,

$$\alpha^{n+1} = \alpha^n + \mathbb{A}\left(\alpha^n, \left(\frac{\mathrm{d}J}{\mathrm{d}\alpha}\right)\right) = \alpha^n + (\Delta\alpha)^n, \tag{1}$$

where $\mathbb{A}$ is an optimization algorithm operator that returns a perturbation $\Delta\alpha$ to the current design $\alpha^n$. The procedure is repeated until a convergence criterion is reached. The design parameters, $\alpha$, are chosen based on the optimization problem and its parametrization.

In CFD applications, computing the term $\frac{\mathrm{d}J}{\mathrm{d}\alpha}$ includes the differentiation of the steady-state partial differential equation (PDE) governing equation of the flow,

$$\boldsymbol{R}(\boldsymbol{\psi}, \alpha) = 0 \rightarrow \frac{\partial \boldsymbol{R}}{\partial \boldsymbol{\psi}} \frac{\partial \boldsymbol{\psi}}{\partial \alpha} + \frac{\partial \boldsymbol{R}}{\partial \alpha} = 0, \tag{2}$$

where $\boldsymbol{R}$ is the residual vector of the discretized flow equations that is driven to zero and $\boldsymbol{\psi}$ stands for state variables (velocity, pressure, temperature, etc.). Eq. (2) results in a linear system,

$$\frac{\partial \boldsymbol{R}}{\partial \boldsymbol{\psi}} \frac{\partial \boldsymbol{\psi}}{\partial \alpha} = -\frac{\partial \boldsymbol{R}}{\partial \alpha}, \tag{3}$$

in which the term $\frac{\partial \boldsymbol{R}}{\partial \boldsymbol{\psi}}$ is Jacobian and $\frac{\partial \boldsymbol{\psi}}{\partial \alpha}$ represents the perturbation of flow fields. Using the chain rule, the total derivative can be then computed by

$$\frac{\mathrm{d}J}{\mathrm{d}\alpha} = \frac{\partial J}{\partial \alpha} + \frac{\partial J}{\partial \boldsymbol{\psi}} \frac{\partial \boldsymbol{\psi}}{\partial \alpha}. \tag{4}$$

Several methods can be used to compute the gradient from Eq. (4). For instance the complex variable technique, which

overcomes the problem of choosing step width in the finite-difference method, or the forward mode (tangent linearization) application of algorithmic differentiation (AD). However, all these methods are computationally expensive when the design space is large. This is due the fact that in Eq. (4) the term $\frac{\partial \psi}{\partial \alpha}$ requires an expensive PDE simulation, which satisfies Eq. (3) for every dimension in the design space, $\alpha_i$. As will be shown in the following, in the adjoint method the sensitivity can be obtained without computing this expensive term.

From Eq. (3) we can write the perturbation term as

$$\frac{\partial \psi}{\partial \alpha} = -\left(\frac{\partial R}{\partial \psi}\right)^{-1} \frac{\partial R}{\partial \alpha} \tag{5}$$

leading to

$$\frac{\mathrm{d}J}{\mathrm{d}\alpha} = \frac{\partial J}{\partial \alpha} - \frac{\partial J}{\partial \psi}\left(\frac{\partial R}{\partial \psi}\right)^{-1} \frac{\partial R}{\partial \alpha} = \frac{\partial J}{\partial \alpha}$$
$$+ \left[-\frac{\partial J}{\partial \psi}\left(\frac{\partial R}{\partial \psi}\right)^{-1}\right] \frac{\partial R}{\partial \alpha}. \tag{6}$$

The terms in the bracket can be identified as the adjoint system of equations from which the adjoint variable, $\overline{\psi}$, can be introduced as

$$\overline{\psi}^T = -\frac{\partial J}{\partial \psi}\left(\frac{\partial R}{\partial \psi}\right)^{-1} \text{ or}$$
$$\left(\frac{\partial R}{\partial \psi}\right)^T \overline{\psi} = -\left(\frac{\partial J}{\partial \psi}\right)^T. \tag{7}$$

In this way, instead of solving a PDE simulation for every design variable, the adjoint system of equations needs to be solved only once. As a result, the computational cost of the gradient becomes independent of the number of design parameters (Giles et al., 2003; Mavriplis, 2007; Nielsen et al., 2010):

$$\frac{\mathrm{d}J}{\mathrm{d}\alpha} = \frac{\partial J}{\partial \alpha} + \overline{\psi}^T \frac{\partial R}{\partial \alpha}. \tag{8}$$

## 3 Derivation of the continuous adjoint solver for ABL inflow calibration

### 3.1 Flow model

The ABL flow model for cases of neutral stratification consists of steady-state Reynolds Averaged Navier–Stokes (RANS) for incompressible fluid flows (Rebollo and Lewandowski, 2014), which results in the following equations for momentum and continuity:

$$(R_1, R_2, R_3)^T = (U \cdot \nabla)U + \nabla p - \nabla \cdot (2\nu_{\mathrm{eff}}\mathbf{D}(U)), \tag{9}$$
$$R_4 = -\nabla \cdot U, \tag{10}$$

where $(R_1, R_2, R_3)$ and $R_4$ denote the discretized flow equations, $R = (R_1, R_2, R_3, R_4)^T$, in Eq. (2). The variables $U$

and $p$ are the state variables velocity vector and modified pressure, $\nu_{\mathrm{eff}}$ stands for the sum of kinematic and turbulent viscosity, and $\mathbf{D}$ is the rate of strain tensor, $\mathbf{D} = \frac{1}{2}(\nabla U + (\nabla U)^T)$. Throughout this work, the standard $k - \epsilon$ turbulence model with the addition of a forest model is used.

### 3.1.1 Inflow boundary

The properties of the inflow boundary have an important effect on the solution of the interior domain. With the increasing application of LES, hybrid RANS-LES and direct numerical simulation (DNS) methods, the inflow turbulence generation has become the subject of many research works in recent decades. For a review of such methods and their application in wind energy, we refer to Wu (2017) and Stevens and Meneveau (2017). However, due to computational cost, it is still popular in the wind energy industry to use the RANS model with an inflow boundary obtained from either an analytical formula or a one-dimensional (1-D) simulation. The idea of the latter method is to solve a zero-pressure gradient equation for a 1-D domain with periodic boundary conditions in the stream- and span-wise directions (Chang et al., 2018):

$$\frac{\partial \overline{U_x' U_z'}}{\partial z} = 0, \tag{11}$$

where $x$ (horizontal) and $z$ (vertical) are the Cartesian coordinates. The inflow profile and its turbulent characteristics are obtained from this 1-D simulation and then are mapped to the 3-D inlet boundary.

### 3.1.2 Forest effect

Forest canopies modify the available free volume of the terrain domain and introduce an explicit drag term to the momentum equation as below:

$$F_{\mathrm{D}} = -\frac{1}{2}\rho C_{\mathrm{d}} A(z)|U|U \tag{12}$$

with density $\rho$, leaf-level canopy drag coefficient $C_{\mathrm{d}}$ and leaf area density $A(z)$. The effects of the forest in the turbulence models such as $k - \epsilon$ for ABL flows have been extensively discussed in the literature and several formulas are presented to make the turbulence model consistent with the modified momentum equation. For more details, the reader may be referred to Lopes da Costa (2007).

### 3.2 Adjoint model

Calibration algorithms seek to maximize the agreement between simulation outputs and measurements. In the context of ABL-based model calibration, the data are often wind speed and direction at one or more locations of a potential wind-farm site. The CFD-based calibration can be formulated as a constrained optimization problem with a scalar objective function:

$$\text{minimize } J(\boldsymbol{U}_{\mathrm{M}}, \boldsymbol{U}_{\mathrm{S}}, \alpha) = \sum_{i=1}^{k} \left[\boldsymbol{U}_{\mathrm{M}_i} - \boldsymbol{U}_{\mathrm{S}_i}\right]^2;$$

$$\text{subject to } \boldsymbol{R}(\boldsymbol{U}, p, \alpha) = 0, \tag{13}$$

where $\boldsymbol{R}$ stands for the spatial residual of the flow equations with $\boldsymbol{U}$ and $p$ the discretized velocity and pressure, respectively. $\boldsymbol{U}_{\mathrm{M}_i}$ and $\boldsymbol{U}_{\mathrm{S}_i}$ are the measured and simulated wind velocities at the same location in the domain, respectively. The variable $\alpha$ represents the design variables which are considered to be the velocity at inlet faces of the CFD mesh through this work.

The derivation of adjoint equations as in the preceding section is arguably the most straightforward way to introduce the adjoint equations and understand their advantages. However, the first developments for using the adjoint equations in CFD applications were done using a Lagrange multiplier argument (Hinze et al., 2008). From this point of view, the inner product of the PDE of flow equations and a new set of variables vanishes the variations of state variables, $\frac{\partial \boldsymbol{U}}{\partial \alpha}$ and $\frac{\partial p}{\partial \alpha}$. By introducing the adjoint variables $\overline{\boldsymbol{U}}$ and $\overline{p}$ for adjoint velocity and adjoint pressure, respectively, $\overline{\boldsymbol{\psi}}^T = (\overline{\boldsymbol{U}}, \overline{p})$, the cost function can be reformulated to a Lagrange function as

$$L := J + \int_{\Omega} \left(\overline{U}_x R_1^T + \overline{U}_y R_2^T + \overline{U}_z R_3^T + \overline{p} R_4^T\right) d\Omega$$

$$= J + \int_{\Omega} (\overline{\boldsymbol{U}}, \overline{p}) \boldsymbol{R} d\Omega, \tag{14}$$

where $\overline{U}_x$, $\overline{U}_y$ and $\overline{U}_z$ are the adjoint velocity components and $\Omega$ is the flow domain. For the sensitivities of the cost function with respect to the design parameters, we have to compute the total variation of $L$:

$$\frac{\mathrm{d}L}{\mathrm{d}\alpha} = \frac{\partial J}{\partial \alpha} + \frac{\partial J}{\partial \boldsymbol{U}} \frac{\partial \boldsymbol{U}}{\partial \alpha} + \frac{\partial J}{\partial p} \frac{\partial p}{\partial \alpha} + \int_{\Omega}$$

$$\left[(\overline{\boldsymbol{U}}, \overline{p}) \left(\frac{\partial \boldsymbol{R}}{\partial \boldsymbol{U}} \frac{\partial \boldsymbol{U}}{\partial \alpha} + \frac{\partial \boldsymbol{R}}{\partial p} \frac{\partial p}{\partial \alpha}\right) + (\overline{\boldsymbol{U}}, \overline{p}) \frac{\partial \boldsymbol{R}}{\partial \alpha}\right] d\Omega. \tag{15}$$

Choosing the Lagrange multipliers such that the variation with respect to the state variables vanishes, leads to

$$\frac{\partial J}{\partial \boldsymbol{U}} \frac{\partial \boldsymbol{U}}{\partial \alpha} + \frac{\partial J}{\partial p} \frac{\partial p}{\partial \alpha} + \int_{\Omega} \left[(\overline{\boldsymbol{U}}, \overline{p}) \left(\frac{\partial \boldsymbol{R}}{\partial \boldsymbol{U}} \frac{\partial \boldsymbol{U}}{\partial \alpha} + \frac{\partial \boldsymbol{R}}{\partial p} \frac{\partial p}{\partial \alpha}\right)\right] d\Omega = 0. \tag{16}$$

Then the sensitivity of the cost function can be given by

$$\frac{\mathrm{d}L}{\mathrm{d}\alpha} = \frac{\partial J}{\partial \alpha} + \int_{\Omega} (\overline{\boldsymbol{U}}, \overline{p}) \frac{\partial \boldsymbol{R}}{\partial \alpha} d\Omega, \tag{17}$$

which excludes the state variable sensitivities.

The theory presented here is based on the work of Othmer (2008), which derives an adjoint solver for topology optimization of duct flows to reduce the pressure loss between inlet and outlet boundaries. A more detailed derivation of the model is given by Hinterberger and Olesen (2011). By neglecting the turbulent viscosity variation, assuming the "frozen turbulence" hypothesis; replacing the derivative of Eq. (9) with the forest source term; and inserting Eq. (10) into Eq. (16), this gives

$$\frac{\partial J}{\partial \boldsymbol{U}} \frac{\partial \boldsymbol{U}}{\partial \alpha} + \frac{\partial J}{\partial p} \frac{\partial p}{\partial \alpha} + \int_{\Omega} \overline{\boldsymbol{U}} \cdot \left[\left(\frac{\partial \boldsymbol{U}}{\partial \alpha} \cdot \nabla\right) \boldsymbol{U} + (\boldsymbol{U} \cdot \nabla) \frac{\partial \boldsymbol{U}}{\partial \alpha}\right.$$

$$- \nabla \cdot \left(2\nu_{\mathrm{eff}} \mathbf{D}\left(\frac{\partial \boldsymbol{U}}{\partial \alpha}\right)\right)$$

$$+ \frac{1}{2} C_D A \left(|\overset{\sim 0}{\frac{\partial \boldsymbol{U}}{\partial \alpha}}|\boldsymbol{U} + |\boldsymbol{U}|\frac{\partial \boldsymbol{U}}{\partial \alpha})\right] d\Omega$$

$$- \int_{\Omega} \overline{p} \nabla \cdot \frac{\partial \boldsymbol{U}}{\partial \alpha} d\Omega + \int_{\Omega} \overline{\boldsymbol{U}} \cdot \nabla \frac{\partial p}{\partial \alpha} d\Omega = 0. \tag{18}$$

Decomposition of parts into interior domain, $\Omega$, and its boundaries, $\Gamma$, leads to reformulation of Eq. (18) as follows

$$\int_{\Gamma} \left[\overline{\boldsymbol{U}} \cdot \boldsymbol{n} + \overset{0}{\frac{\partial J_{\Gamma}}{\partial p}}\right] \frac{\partial p}{\partial \alpha} d\Gamma + \int_{\Gamma} \left[\boldsymbol{n}(\overline{\boldsymbol{U}} \cdot \boldsymbol{U}) + \overline{\boldsymbol{U}}(\boldsymbol{U} \cdot \boldsymbol{n})\right.$$

$$+ 2\nu_{eff}\boldsymbol{n} \cdot \mathbf{D}(\overline{\boldsymbol{U}}) - \overline{p}\boldsymbol{n} + \overset{0}{\frac{\partial J_{\Gamma}}{\partial \boldsymbol{U}}}\right] \frac{\partial \boldsymbol{U}}{\partial \alpha} d\Gamma$$

$$+ \int_{\Gamma} \left[-2\nu_{\mathrm{eff}}\boldsymbol{n} \cdot \mathbf{D}\left(\frac{\partial \boldsymbol{U}}{\partial \alpha}\right) \cdot \overline{\boldsymbol{U}}\right] d\Gamma$$

$$+ \int_{\Omega} \left[-\nabla \cdot \overline{\boldsymbol{U}} + \overset{0}{\frac{\partial J_{\Omega}}{\partial p}}\right] \frac{\partial p}{\partial \alpha} d\Omega$$

$$+ \int_{\Omega} \left[-\nabla \overline{\boldsymbol{U}} \cdot \boldsymbol{U} - (\boldsymbol{U} \cdot \nabla)\overline{\boldsymbol{U}} - \nabla \cdot \left(2\nu_{\mathrm{eff}}\mathbf{D}(\overline{\boldsymbol{U}})\right)\right.$$

$$+ \frac{1}{2} C_D A |\boldsymbol{U}|\overline{\boldsymbol{U}} + \nabla \overline{p} + \frac{\partial J_{\Omega}}{\partial \boldsymbol{U}}\right] \frac{\partial \boldsymbol{U}}{\partial \alpha} d\Omega = 0. \tag{19}$$

$J_{\Gamma}$ and $J_{\Omega}$, respectively, stand for the part of the cost function which is dependent on the flow state values on the boundary and volume of the domain. Due to the definition of the cost function (Eq. 13), its direct variation comes only from the interior domain $\left(\frac{\partial J_{\Gamma}}{\partial \boldsymbol{U}} = 0, \frac{\partial J_{\Gamma}}{\partial p} = 0\right)$. Moreover, it does not have any derivative of the pressure field $\left(\frac{\partial J_{\Omega}}{\partial p} = 0\right)$. The only derivative of the cost function is with respect to the inflow and velocity in the interior of the domain and at the locations

where the measurements are available. From the latter we have

$$\frac{\partial J_{\Omega}}{\partial \boldsymbol{U}} = -2\left(\boldsymbol{U}_{M_i} - \boldsymbol{U}_{S_i}\right) \quad i = 1, 2, \dots. \tag{20}$$

Using Eq. (19) and (20) the adjoint equations can be derived as

$$-\nabla \overline{\boldsymbol{U}} \cdot \boldsymbol{U} - (\boldsymbol{U} \cdot \nabla)\overline{\boldsymbol{U}} = -\nabla \overline{p} + \nabla \cdot \left(2\nu_{\text{eff}}\mathbf{D}(\overline{\boldsymbol{U}})\right)$$
$$+ \left(\frac{2}{\omega_i}\right)\left(\boldsymbol{U}_{M_i} - \boldsymbol{U}_{S_i}\right) - \frac{1}{2}C_{\text{D}}A|\boldsymbol{U}|\overline{\boldsymbol{U}}, \tag{21}$$

$$\nabla \cdot \overline{\boldsymbol{U}} = 0, \tag{22}$$

where $\omega_i$ is the volume of the cell in which the measurement is located.

### 3.2.1 Boundary conditions

The boundary integrals of Eq. (19) can be mathematically reformulated and reduced to

$$\int_{\Gamma} [\overline{\boldsymbol{U}} \cdot \boldsymbol{n}]\frac{\partial p}{\partial \alpha}d\Gamma = 0, \tag{23}$$

$$\int_{\Gamma} [\boldsymbol{n}(\overline{\boldsymbol{U}} \cdot \boldsymbol{U}) + \nu_{\text{eff}}(\boldsymbol{n} \cdot \nabla)\overline{\boldsymbol{U}} - \overline{p}\boldsymbol{n}] \cdot \frac{\partial \boldsymbol{U}}{\partial \alpha}d\Gamma$$
$$- \int_{\Gamma} \left[\nu_{\text{eff}}(\boldsymbol{n} \cdot \nabla)\frac{\partial \boldsymbol{U}}{\partial \alpha} \cdot \overline{\boldsymbol{U}}\right]d\Gamma = 0, \tag{24}$$

where $\boldsymbol{n}$ is the unit normal vector from the boundary faces. Except for the inlet, which is the design space, the adjoint BCs should be chosen such that the above equations are held.

Generally, for an ABL CFD domain no-slip wall (zero fixed velocity) and zero-pressure gradient conditions are imposed on the ground. For a wall type of boundary in which $\frac{\partial \boldsymbol{U}}{\partial \alpha}$ is zero, the first integral of Eq. (24) is canceled. Then, the only way to satisfy the conditions,

$$\overline{\boldsymbol{U}} \cdot \boldsymbol{n} = 0, \tag{25}$$

$$(\boldsymbol{n} \cdot \nabla)\frac{\partial \boldsymbol{U}}{\partial \alpha} \cdot \overline{\boldsymbol{U}} = 0, \tag{26}$$

is to apply a no-slip ($\overline{\boldsymbol{U}} = 0$) condition on the ground. No BC can be derived on the ground for the adjoint pressure but consistent with the primal a zero gradient condition is applied.

For the top and outlet boundaries of the domain a zero gradient velocity ($(\boldsymbol{n} \cdot \nabla)\frac{\partial \boldsymbol{U}}{\partial \alpha} = 0$) and zero fixed pressure ($p = 0$) are the common conditions for the primal system. These conditions fulfill Eq. (23) and cancel the second integral of Eq. (24). The only term that remains is the first term of Eq. (24), which needs to be zeroed out,

$$\left[\boldsymbol{n}(\overline{\boldsymbol{U}} \cdot \boldsymbol{U}) + \nu_{\text{eff}}(\boldsymbol{n} \cdot \nabla)\overline{\boldsymbol{U}} - \overline{p}\boldsymbol{n}\right] \cdot \frac{\partial \boldsymbol{U}}{\partial \alpha} = 0. \tag{27}$$

After decomposition into tangent and normal components it can be shown that the relations below should hold

$$\overline{p} = \overline{\boldsymbol{U}} \cdot \boldsymbol{U} + U_n \overline{U}_n + \nu_{\text{eff}}(\boldsymbol{n} \cdot \nabla)\overline{U}_n, \tag{28}$$

$$0 = U_n \overline{U}_t + \nu_{\text{eff}}(\boldsymbol{n} \cdot \nabla)\overline{U}_t, \tag{29}$$

where subscripts $n$ and $t$ represent the normal and in-plane components, respectively. The adjoint BCs can be summarized as

$$\text{ground (wall)} : \overline{\boldsymbol{U}} = 0, \quad \boldsymbol{n} \cdot \nabla \overline{p} = 0; \tag{30}$$

$$\text{top/outlet} : \overline{p} = \overline{\boldsymbol{U}} \cdot \boldsymbol{U} + U_n \overline{U}_n + \nu_{\text{eff}}(\boldsymbol{n} \cdot \nabla)\overline{U}_n,$$
$$\overline{U}_t = 0. \tag{31}$$

It is worth mentioning that the last term of the adjoint pressure, which includes the kinematic viscosity, in implementation is often neglected (Nilsson et al., 2014). Moreover, the adjoint variables at the inlet should not be chosen to zero out the inlet velocity perturbations because the design variables are the inlet velocities. Instead, the zero gradient condition is imposed on the inlet for both adjoint velocity and adjoint pressure to have a well-posed system. Finally, from the integral over the boundary term in Eq. (19) it is clear one needs to evaluate the following expression,

$$\frac{\partial J}{\partial \alpha} = \frac{\partial J}{\partial \boldsymbol{U}_{\text{inlet}}} = \boldsymbol{n}\left(\overline{\boldsymbol{U}}_{\text{inlet}} \cdot \boldsymbol{U}_{\text{inlet}}\right)$$
$$+ \overline{\boldsymbol{U}}_{\text{inlet}}(\boldsymbol{U}_{\text{inlet}} \cdot \boldsymbol{n}) + 2\nu_{\text{eff}}\boldsymbol{n} \cdot \mathbf{D}\left(\overline{\boldsymbol{U}}_{\text{inlet}}\right), \tag{32}$$

to compute the sensitivity.

### 3.2.2 Wind direction effect

As it was mentioned before, in ABL CFD simulations it is common to simulate first a 1-D domain with a periodic boundary to obtain the inflow boundary condition. Then the cell center velocity of the 1-D run is copied directly to its counterpart boundary face in the 3-D domain. As a requirement, the number of cells in the 1-D mesh and faces in the vertical direction of the 3-D inflow boundary should be the same (see Fig. 1). Moreover, and ideally, the face center heights in the 3-D mesh are equal to their counterpart cell height in 1-D. Although, in current work, a circular inflow–outflow boundary is considered, with some small modification in the code the method can also be applied to other boundary shapes.

The inflow wind direction (WD) effect can be expressed by a rotation angle, $\theta$, which rotates the inflow from its default west-to-east (WD = 270°) direction,

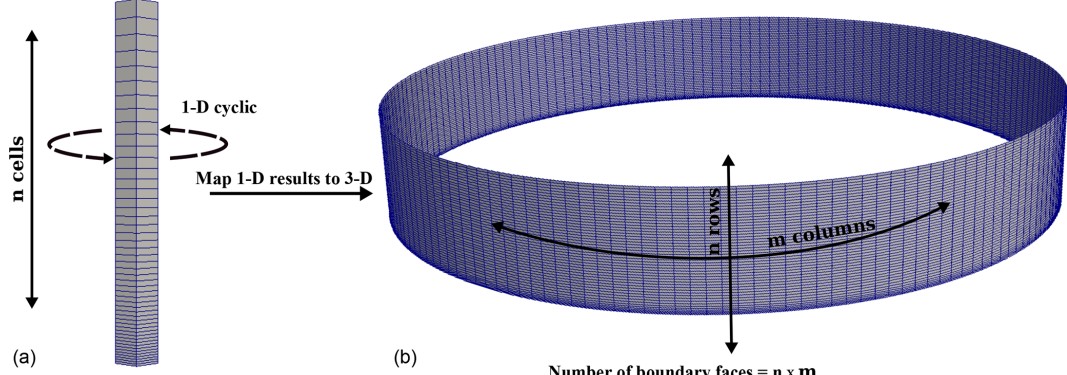

**Figure 1.** The inflow velocities of each cell from 1-D precursor run (**a**) are copied to the boundary of the 3-D domain (**b**), which has a similar number of cells in the vertical direction. Ideally the height of each face in the 3-D domain boundary is exactly the same as its counterpart cell in the 1-D mesh.

$$WD = 270° − \theta, \tag{33}$$

$$(\boldsymbol{U}_{\text{inlet}})_x = \boldsymbol{U}_{\text{1-D}} \cos(\theta), \tag{34}$$

$$(\boldsymbol{U}_{\text{inlet}})_y = \boldsymbol{U}_{\text{1-D}} \sin(\theta), \tag{35}$$

$$(\boldsymbol{U}_{\text{inlet}})_z = 0. \tag{36}$$

The differentiation of Eqs. (34) and (35) gives

$$\frac{\partial (\boldsymbol{U}_{\text{inlet}})_x}{\partial \boldsymbol{U}_{\text{1-D}}} = \cos(\theta); \quad \frac{\partial (\boldsymbol{U}_{\text{inlet}})_x}{\partial \theta} = −\boldsymbol{U}_{\text{1-D}} \sin(\theta); \tag{37}$$

$$\frac{\partial (\boldsymbol{U}_{\text{inlet}})_y}{\partial \boldsymbol{U}_{\text{1-D}}} = \sin(\theta); \quad \frac{\partial (\boldsymbol{U}_{\text{inlet}})_y}{\partial \theta} = \boldsymbol{U}_{\text{1-D}} \cos(\theta). \tag{38}$$

The adjoint solver which was explained in the previous section computes the derivative of the cost function with respect to 3-D inflow velocities at each face of the boundary:

$$\frac{\partial J}{\partial (\boldsymbol{U}_{\text{inlet}})_x}\Big|_{i,j}; \quad \frac{\partial J}{\partial (\boldsymbol{U}_{\text{inlet}})_y}\Big|_{i,j}; \; i = 1, \ldots, n; \; j = 1, \ldots, m, \tag{39}$$

where $i$ and $j$ represent the row- and column-wise position of a face on the boundary and the total number of the faces in the 3-D circular boundary is $N = n \times m$.

Using the chain rule, one can compute the sensitivity with respect to each cell of the 1-D inflow velocity as follows

$$\begin{aligned}
\frac{\partial J}{\partial \boldsymbol{U}_{\text{1-D}}}\Big|_i &= \left(\sum_{j=1}^{m} \frac{\partial J}{\partial (\boldsymbol{U}_{\text{inlet}})_x}\Big|_{i,j}\right) \frac{\partial (\boldsymbol{U}_{\text{inlet}})_x}{\partial \boldsymbol{U}_{\text{1-D}}} \\
&+ \left(\sum_{j=1}^{m} \frac{\partial J}{\partial (\boldsymbol{U}_{\text{inlet}})_y}\Big|_{i,j}\right) \frac{\partial (\boldsymbol{U}_{\text{inlet}})_y}{\partial \boldsymbol{U}_{\text{1-D}}} \\
&= \left(\sum_{j=1}^{m} \frac{\partial J}{\partial (\boldsymbol{U}_{\text{inlet}})_x}\Big|_{i,j}\right) \cos(\theta) \\
&+ \left(\sum_{j=1}^{m} \frac{\partial J}{\partial (\boldsymbol{U}_{\text{inlet}})_y}\Big|_{i,j}\right) \sin(\theta).
\end{aligned} \tag{40}$$

Eq. (40) means the $x$ and $y$ gradient components of the 3-D inflow faces at the same column $j$ are accumulated and then

multiplied by $\cos(\theta)$ and $\sin(\theta)$, respectively, before being summed. For the sake of clarity, Eq. (40) can be rewritten as

$$\frac{\partial J}{\partial \boldsymbol{U}_{\text{1-D}}} = \frac{\partial J}{\partial (\boldsymbol{U}_{\text{3-D}})_x} \cos(\theta) + \frac{\partial J}{\partial (\boldsymbol{U}_{\text{3-D}})_y} \sin(\theta). \tag{41}$$

Using the same analogy, it can be shown that the sensitivity with respect to rotation angle can be obtained by

$$\begin{aligned}
\frac{\partial J}{\partial \theta} &= −\left[\frac{\partial J}{\partial (\boldsymbol{U}_{\text{3-D}})_x} \cdot \boldsymbol{U}_{\text{1-D}}\right] \sin(\theta) \\
&+ \left[\frac{\partial J}{\partial (\boldsymbol{U}_{\text{3-D}})_y} \cdot \boldsymbol{U}_{\text{1-D}}\right] \cos(\theta),
\end{aligned} \tag{42}$$

where the dot sign stands for the inner product of the two vectors.

## 4   Numerical results

The adjoint solver and Eq. (32) are implemented based on the "*simpleFoam*" incompressible CFD solver of OpenFOAM-4.1. In this section, first the accuracy of the gradients obtained by the developed solver is tested against the 2nd-order FD method in a simple 3-D domain. Then, the inflow calibration of a real complex terrain is presented. For all the simulations in this work the general roughness length of the domain is $z_0 = 0.05$ (m) and the turbulent eddies are modeled by standard $k − \epsilon$ model with canopy model by Liu et al. (1996). Moreover, an ABL wall function is used to apply the roughness-related logarithmic law near to the ground, which is consistent with Monin–Obukhov similarity theory (Chang et al., 2018).

### 4.1   Gradient verification

A cylindrical domain with 1000 m radius and 300 m height is chosen. The mesh of the domain has 209 000 hexahedral cells in which the inflow–outflow boundary has 49 rows

**Table 1.** Comparison of the wind direction gradients by finite differences (FD) and via the adjoint approach.

|  | Adjoint | Finite difference | $\epsilon_{\text{rel}}$ |
|---|---|---|---|
| $\frac{\mathrm{d}J}{\mathrm{d}\theta}$ | 215 | 202 | 0.06 |

and 172 columns and a total number of 8428 faces ($N = 49 \times 172 = 8428$). The *topoSet* utility of OpenFOAM is used to select a number of cells as forest in a box size of 300 m in $x$ and $y$ and 40 m in $z$ direction. The canopy drag and leaf area density for all forest cells are $C_{\mathrm{d}} = 0.3$ and $A = 0.0033\,\mathrm{m}^{-1}$, respectively. The operational Reynolds number based on the free-stream velocity, forest area height and air kinematic viscosity is $Re_{\mathrm{h}} = 4.8 \times 10^5$.

For the primal CFD simulation of a circular domain the standard "*inletOutlet*" BC is used which checks if the flow is flowing into the domain or out of it and switches between fixed value and zero gradient, respectively. This BC is further developed to apply the derived adjoint BCs in a similar way and based on the flow direction on the boundary.

The gradient evaluation is carried out for a simulation in which the inflow wind direction is WD = 240°. To have some reference wind speeds the terrain is simulated assuming the wind blows from west to east (WD = 270°). The domain and the velocity field on the planes $y = 0\,\mathrm{m}$ and $z = 45\,\mathrm{m}$ are shown in Fig. 2. The 1-D inflow boundary and the target velocity profiles in the domain are plotted in Fig. 3. The forest effect can be seen in the flow field and on the wind shear of the profile. Please note that the 30° difference between these two simulations is not the step size for the finite-difference computation. The finite-difference step size for gradient validation of wind direction is 0.3°.

The gradients obtained by the developed adjoint solver are plotted against the 2nd-order FD gradients in Fig. 4. In general, the trends of the sensitivity profiles are similar. Moreover, the gradients are in excellent agreement except only for the heights between 50 and 100 m in which the maximum relative error is $\epsilon_{\text{rel}} = 0.10$. This difference can be traced back to grid resolution and the derivation of the adjoint equations and BCs which includes some simplifications such as the assumption of the frozen turbulence.

The wind direction gradients are tabulated in Table 1. The derivative of calibration cost function with respect to the change in wind direction is much higher than the change in the inflow boundary. Here also, there is a good agreement between the FD and the adjoint gradients and the relative error of wind direction sensitivity is close to that of inflow gradients. This is of course not surprising because it was shown in the previous section the sensitivity with respect to $\theta$ is obtained by mathematical operations only after the adjoint gradients are available. That said, the accuracy of the gradients computed by the adjoint solver and their signs show that they can be used for the purpose of the gradient-based calibration.

## 4.2 Inflow calibration

For the cases, studied in this section, the in-house *terrainMesher* software of the Fraunhofer IWES is used to generate the mesh. The primal flow field is simulated with an in-house CFD solver (Chang et al., 2018). The solver is a customized ABL-based version of the *simpleFoam* solver in the OpenFOAM package with a modified $k - \epsilon$ turbulence model which behaves like a standard model for the neutral condition.

To optimize the inlet velocity profile, the primal and adjoint solvers are coupled with the DAKOTA optimization package (Adams et al., 2017). The local gradient-based "CONMIN-frcg" solver of DAKOTA is used, which is based on the conjugate-gradient algorithm by Fletcher and Reeves (Reeves and Fletcher, 1964; Hager and Zhang, 2006). Starting from an initial guess, the algorithm updates the design parameters, $\alpha^n$, using the recurrence of Eq. (1) in which

$$(\Delta\alpha)^n = s^n \mathbb{D}^n \tag{43}$$

and the positive step size $s^n$ is obtained by a line search, and the directions $\mathbb{D}$ are generated by the following rules:

$$\mathbb{D}^{n+1} = -g^{n+1} + \beta^n \mathbb{D}^n; \quad g^n = \left[\nabla J\left(\alpha^n\right)\right]^T;$$

$$\beta^n = \frac{\left|g^{n+1}\right|^2}{\left|g^n\right|^2}; \quad \mathbb{D}^0 = -g^0. \tag{44}$$

Figure 5 shows the flow chart of the calibration, in which all the steps are followed sequentially. The optimizer starts with an initial guess (both inlet velocity and WD) and repeatedly asks either for the cost function value or the gradients. The primal and the adjoint solvers provide the required information, whenever it is needed, and this process continues until a certain convergence criterion is satisfied.

As a simplistic method, the inflow boundary of a RANS ABL domain can be represented by an empirical power law function or an analytically obtained logarithmic function, which is based on the Monin–Obukhov similarity theory (MOST) (Foken, 2006). During optimization, the optimizer may ask for a cost function evaluation with a new inflow boundary, which is highly unrealistic for an ABL domain, leading to poor numerical stability or even divergence. One may assume that the inflow boundary is an analytical empirical function and, instead of the inflow velocities, calibrate the parameters of that function. Having the gradient via adjoint solver and using the chain rule, the gradient of the cost function with respect to these parameters can be easily obtained. However, the inflow boundary of a real 3-D complex terrain is neither a power law nor a logarithmic function and such parametrization may fail.

A simple approach is used in this study. This part is called the feasibility check in the optimization flow chart and is a Python script. First of all, the optimizer output is smoothed to avoid having any spikes in the inflow profile. Then it is

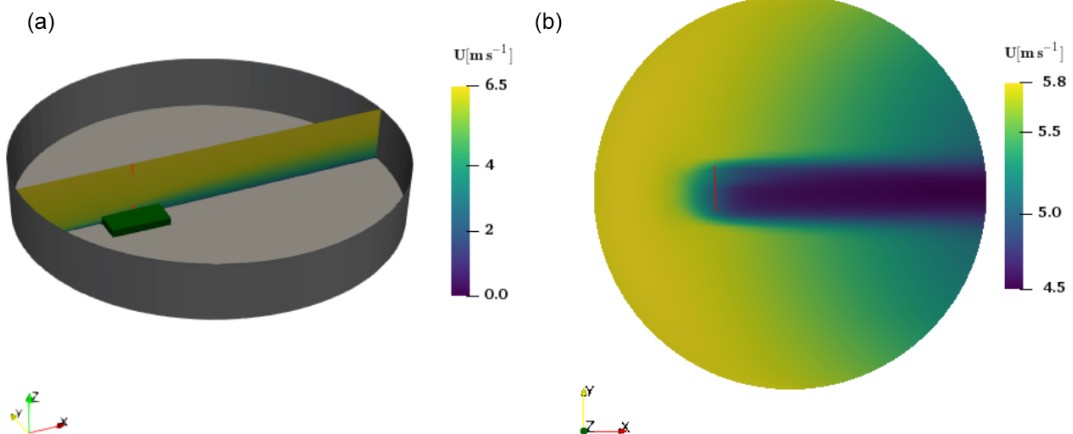

**Figure 2.** The CFD-simulated velocity obtained from the reference simulation (WD $= 270°$): the general view of the domain including the cubic forest and the velocity field on a plane at the center of the domain, $y = 0$, **(a)**. The red line on the plane represents the location of desired target velocity profile (see Fig. 3). The velocity field on a plane at $z = 45$ m above the ground **(b)**.

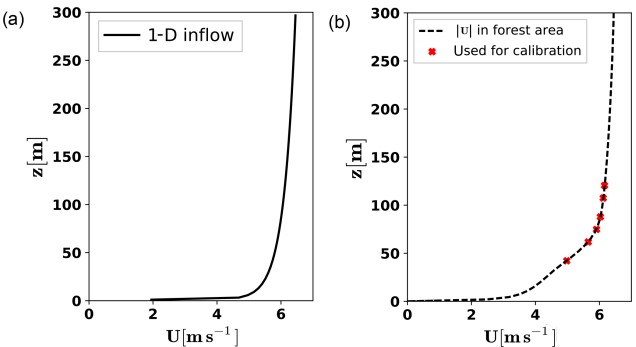

**Figure 3.** The 1-D inflow boundary **(a)** and the velocity profile and its selected target speeds in the domain **(b)**.

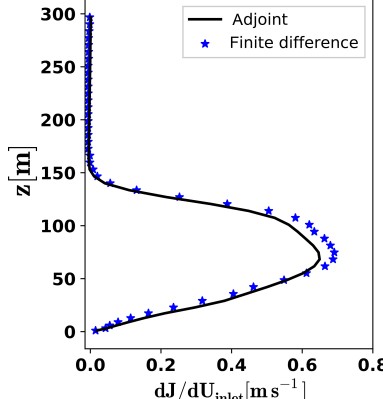

**Figure 4.** Comparison of the inflow boundary gradients by finite differences (FD) and via the adjoint approach.

checked whether a logarithmic, $f_1(x) = A \ln(Bx + C)$, or a power law, $f_2(x) = A(\frac{x}{B})^C$, function could be fitted into it. If either of these functions is fitted and its coefficient of determination is above 0.96, the smoothed inflow from optimizer (not the fitted profile!) is accepted for the CFD solver. Otherwise, the optimization takes the last fitted profile and asks for a new gradient evaluation. In this way, the inflow boundary is not necessarily a logarithmic or power law profile, and, moreover, it is not so unrealistic to be problematic for the solver. As an alternative, constraints or a penalization term can be added to the objective function. This will be explored in future works when for instance the inflow turbulence properties are also considered design parameters.

### 4.2.1 Ishihara case

As a case study, an ABL domain with a 3-D hill at the center is considered (see Fig. 6). The hill has the shape $z = h \cos^2(\sqrt{x^2 + y^2}/2L)$ with $h = 40$ m and $L = 100$ m. The scaled wind tunnel study of the case has been presented by Ishihara et al. (1999). The domain is meshed with $2 \times 10^6$ hexahedral elements. The roughness value of the domain is set to be $z_0 = 0.04$ m. The operational Reynolds number based on the free-stream velocity, hill height and air kinematic viscosity is $Re_h = 1.5 \times 10^4$.

Using Ishihara et al. (1999) wind tunnel measurements, the $x$ component of the velocity over the hill ($U_x$) is used for the inflow velocity calibration. The velocity flow field at the center of the domain and the wake behind the hill can be seen in Fig. 6. Both the primal and the adjoint solvers have parallel scalability of OpenFOAM toolbox. The runtime of the adjoint solver is 60 %–70 % of the primal flow simulation, which has 30 min wall-clock time run with 24 CPU cores. It is worth noting that in the adjoint solver there is no adjoint turbulence equation to be solved.

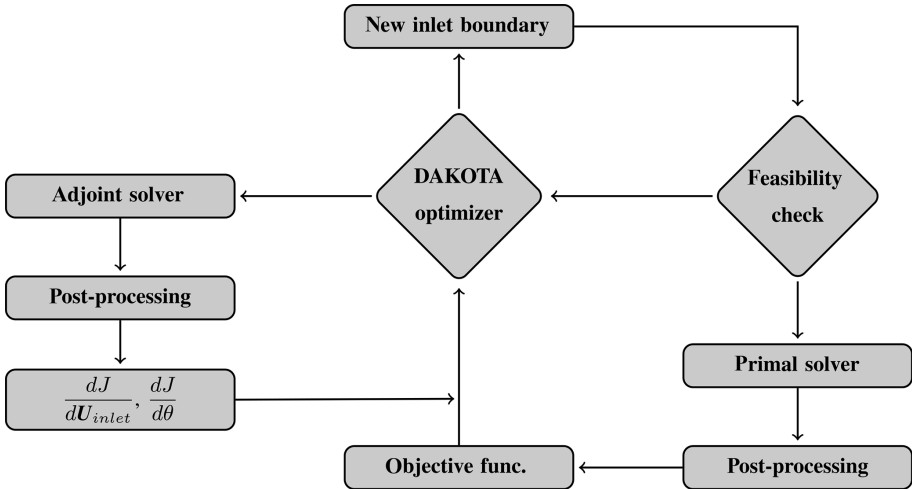

**Figure 5.** The calibration flowchart of the inflow boundary using the DAKOTA optimization package and the developed adjoint solver.

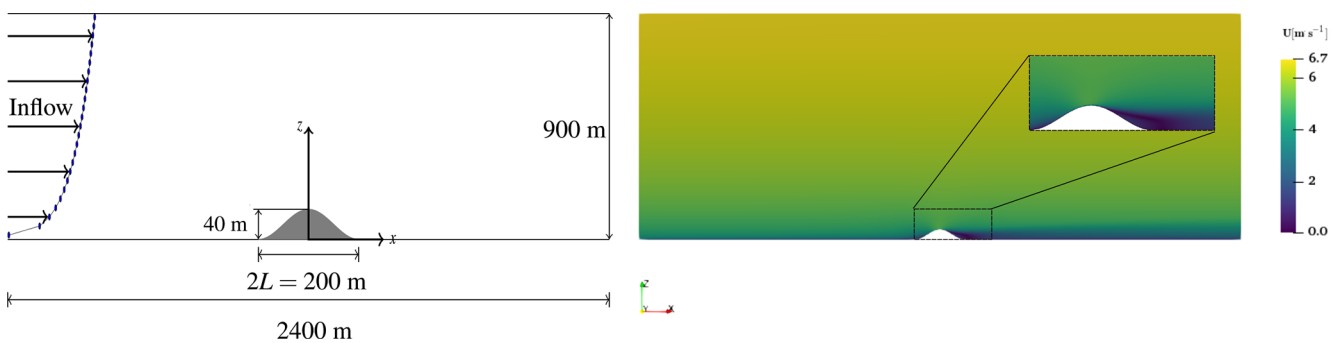

**Figure 6.** Main dimensions of the test case with 1000 m length in the $y$ direction. The velocity flow field on the plane at the center of the domain, $y = 0$.

The stopping criterion is such that the absolute error between the measurements and the simulated velocities over the hill is $\epsilon_{\mathrm{abs}} < 0.1\,\mathrm{m\,s^{-1}}$. Figure 7 shows the history of optimization and the comparison of inlet velocity profiles. The optimization has converged with 14 primal and 12 adjoint calls. The optimal velocity profile is in good agreement with the experiment. There is a small deviation starting from the height $\frac{z}{h} > 2$. However, the comparison of the normalized velocity profiles over the hill, shown in Fig. 8, confirms that the optimized inlet boundary is able to reproduce the experiment velocity profiles.

### 4.2.2 Kassel case

For the second case study, the neutral condition of "Kassel Experiment" is considered. The domain, located near Kassel in Germany, is one of the cases of the New European Wind Atlas (NEWA) project (EU-ERA-NET, Accessed September 9, 2018,). There are two meteorological masts at the site including a 200 m high mast (MM200). The wind rose of the site, provided by NEWA, indicates that most of the time the wind blows from the southwest (SW) to northeast (NE).

The site is represented by a cylindrical domain with 15 km radius and 4 km height (see Fig. 9). The structured mesh is generated with the Fraunhofer IWES in-house software "*terrainMesher*" with 80 cells in the vertical direction. A mesh independence study is conducted to verify the suitability of a mesh of $7 \times 10^6$ hexahedral elements. The vicinity of the hill ($z_{\mathrm{hill}} = 428\,\mathrm{m}$) where the MM200 mast is installed consists of forested area. The leaf area density of the site is obtained from the airborne lidar data and is provided by the NEWA project (Dörenkämper, 2018).

Although the wind speed measurements of the MM200 mast from the site are available, there is not enough information for wind direction at certain heights. This information is necessary for the adjoint solver in which the difference between the components of the measured and simulated velocities is a force term on the right-hand side of the adjoint momentum equation (Eq. 21). Moreover, in the calibration of the solver with real measurements, it would become difficult to discuss the source of error when there is a discrepancy between the target and the calibrated profiles. This is because, aside from the inflow calibration process, which is the aim

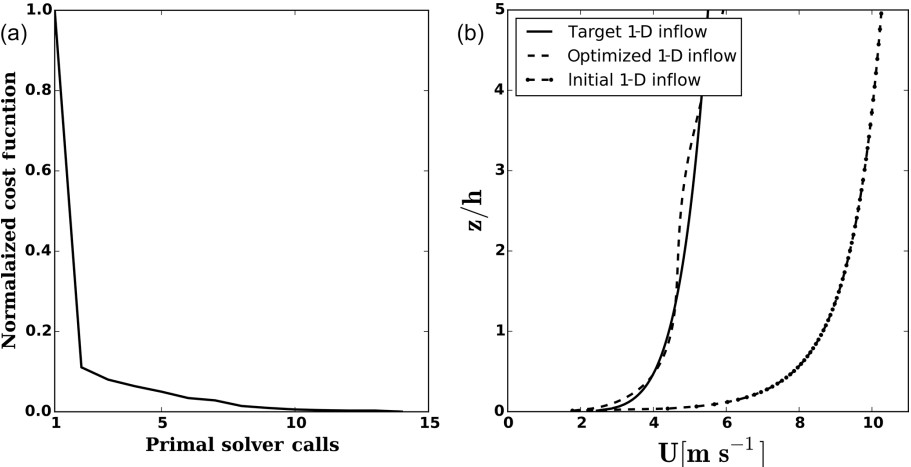

**Figure 7.** Optimization history **(a)** and inlet velocity profile comparison **(b)**.

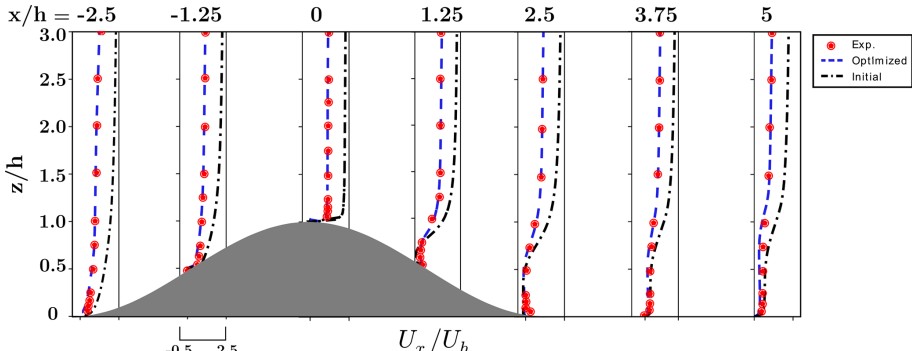

**Figure 8.** Normalized vertical profiles of longitudinal velocity component on the central plane of the hill. $U_h$ is the velocity at the hill height in the undisturbed region of the domain between inlet and hill.

of the current study, many other parameters (e.g., turbulence model and the accuracy of forest and ground roughness map) are involved. Instead of using the real measurements of the mast, the velocity profile near to the mast from a reference simulation is considered. The selected wind speeds can be regarded as some pseudo-measurements.

The initial guess WD is defined as 270°, meaning wind blows from west to east. The turbulent properties of the inflow boundary are not part of the design parameters; but instead, in each new flow solver call of the optimization, the turbulent parameters of the $k - \epsilon$ model inside of the domain are initialized with the last converged solution. In this way, the turbulence model parameters are also gradually updated toward the end of the optimization when the inlet boundary velocities have reached their optimum value. The adjoint solver runtime for this case is also 60 %–70 % of the primal solver wall-clock time, which is 33 min with 120 CPU cores.

The convergence criterion is defined such that the optimization stops when the absolute difference between simulated and measured value, $\epsilon_{abs}$, at all heights is below a certain value. Figure 10 shows the history of optimization

and gradients. The optimization has called for a total number of 41 primal solver runs for cost function evaluation. In addition eight adjoint gradient evaluations were required. The optimization convergence graph shows that there is a spike both in the cost function and the wind direction. This can be explained by the fact that in early iterations the derivative of cost function with respect to the WD is much bigger than with respect to inflow. The optimizer updates the WD based on the first gradient computation. This continues for a few iterations until the cost function value, instead of decreasing, increases. Then the optimizer calls for a new gradient computation. From that point onward both the inlet velocities and the wind direction are gradually updated. The optimum WD is found as 216°.

The initial and optimal results are compared in Fig. 11. The calibration is stopped based on a criterion, which is defined as $\epsilon_{abs} < 0.2 \, \mathrm{m \, s^{-1}}$. Although there is a very small deviation between target and optimized inflow profiles, the velocity profile near to the MM200 mast is in good agreement with the pseudo-measurements at all five selected heights and its error is well below the accepted threshold. Here a couple

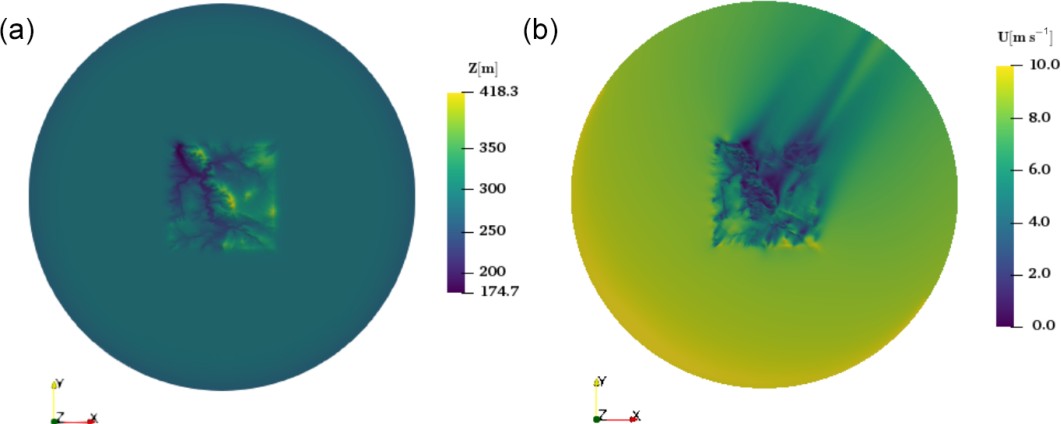

**Figure 9.** The cylindrical domain of the Kassel terrain (a) and the velocity field at the site at 40 m perpendicular distance from the terrain's surface points, which is simulated with WD = 213° (b). This simulation was used as the reference for the calibration.

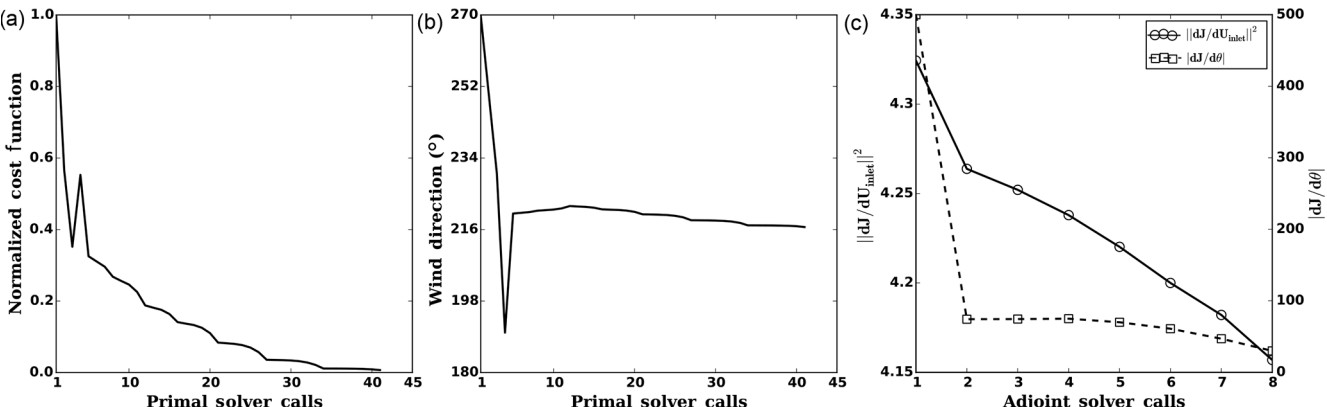

**Figure 10.** Inflow boundary calibration for the Kassel domain; cost function convergence (a), wind direction history (b) and gradients history (c).

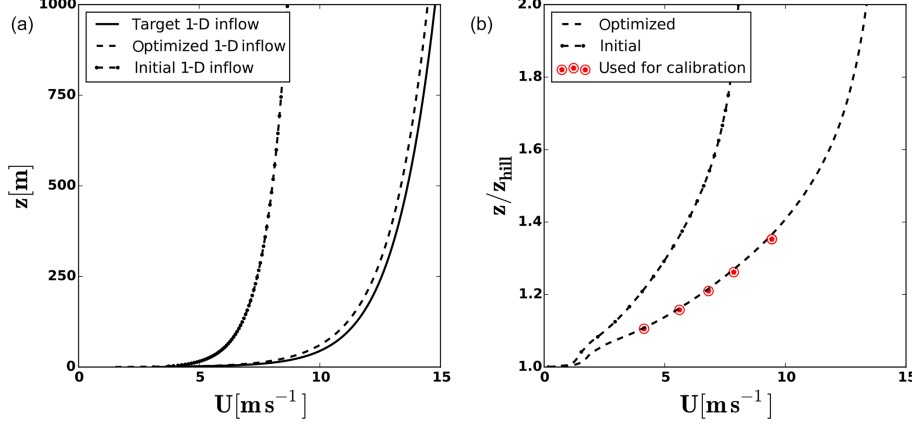

**Figure 11.** The target, initial and optimized 1-D inflow velocity (a). The velocity profile over the hill from which the pseudo-measurements are chosen (b).

of points should also be noted. Firstly, a tighter error criterion would increase the calibration iterations and subsequently the number of CFD solver calls. Secondly, the sensitivity of wind speed at a certain point in the domain with respect to a very small change in the inflow boundary is dependent on many parameters such as terrain complexity, wind direction, CFD model, etc., and cannot be easily generalized.

## 5  Conclusions

In this paper, it has been shown that the ABL CFD solvers can be calibrated via adjoint-based inlet boundary optimization. Based on the frozen turbulence hypothesis and using the reference wind speeds at certain heights inside of the domain, the adjoint equations and its boundary conditions for such problems are derived. The developed solver has been coupled with the DAKOTA optimization package and applied to two 3-D terrains for a neutral stratification condition: (1) the Ishihara test case and (2) the Kassel case. For the Ishihara test case, the optimal inflow was reached with 14 primal and 12 adjoint solver calls. The calibration of the inflow and its directions for the complex terrain of Kassel was achieved after 41 primal and 8 adjoint solver calls. In both cases, an absolute error between the measurements and the simulated velocities was used as a stopping criterion.

The main conclusion remarks of the study can be summarized as follows. (a) The developed calibration framework and the adjoint solver can be successfully applied to even complex domains. (b) The feasibility check of the optimizer output is crucial. Otherwise, at some point in the calibration process, the requested inflow leads to either poor performance or even complete failure (i.e., divergence) of the CFD solver. (c) The convergence criterion can have a big impact on the total number of solver calls. One possibility is to associate the criterion with the uncertainty of the measurements, which can be explored in future work. (d) The process can be further improved to reduce the number of CFD solver calls. For instance, a quasi-Newton optimizer (e.g., BFGS, the Broyden–Fletcher–Goldfarb–Shanno algorithm) or a better parametrization could be used. (e) The presented adjoint solver has the potential to be further developed by including Coriolis force, turbulence model and thermal stratification.

**Data availability.** The forested hill Kassel of Rödeser Berg in Germany is one of the cases of the New European Wind Atlas (NEWA) project (https://map.neweuropeanwindatlas. eu/, last access: November 2019). The "Kassel Experiment" data used in this work are available at https://windbench.net/ newa-r-deser-berg-2017-blind-test (last access: November 2019).

**Author contributions.** SA conceived the presented idea, developed the theory and implemented it. HK generated the mesh and provided the settings for primal flow simulations. RB helped carry out the simulations and post-processing of the results. GS and BS supervised the work. All authors discussed the results and contributed to the final article.

**Competing interests.** The authors declare that they have no conflict of interest.

**Acknowledgements.** The work presented in this paper was part of the ETESIAN (FKZ 0324000D) project. The ETESIAN project was funded by the German Federal Ministry for Economic Affairs and Energy due to a decision of the German Bundestag. Moreover, the Kassel domain, which was used for the numerical results, is a test case from the NEWA project. The NEWA project was funded by the German Federal Ministry for Economic Affairs and Energy (ref. no. 0325832A/B) on the basis of a decision by the German Bundestag with further financial support from NEWA ERA-NET Plus, topic FP7-ENERGY.2013.10.1.2. The simulations were performed at the high-performance computing cluster EDDY, located at the University of Oldenburg (Germany), and funded by the Federal Ministry for Economic Affairs and Energy (Bundesministerium für Wirtschaft und Energie) under grant number 0324005.

**Financial support.** This research has been supported by the German Federal Ministry for Economic Affairs and Energy (grant nos. FKZ0324000D, 0324005 and 0325832A/B) and the German Bundestag with further financial support from NEWA ERA-NET Plus (grant no. FP7-ENERGY.2013.10.1.2).

**Review statement.** This paper was edited by Raúl Bayoán Cal and reviewed by four anonymous referees.

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
