# Peer review of "Adjoint-based Calibration of Inlet Boundary Condition for Atmospheric CFD Solvers"

_Wind Energy Science, 2019_

## Referee Comment (RC1) · Anonymous Referee #1 · 4 Mar 2019

In the manuscript the authors introduce an adjoint based calibration method that can be used to ensure that the inflow boundary conditions represent some predetermined inflow characteristics. For comparison to field measurement data a correct representation of the inflow turbulent boundary layer properties can be very important and the present method can improve this. The presented methods thus seems to be useful for wind turbine or wind farm simulations and will therefore be interesting for readers of the journal.

After reading the manuscript I have various suggestions and comments, which I hope can improve the description of the work

a. Page 2 states "Mainly two different methods are used to set the inflow boundary conditions for ABL flow simulations". There are various other methods that are used within the community such as using white noise, Mann spectrum, (concurrent) precursor methods, etc. For a detailed discussion of turbulence inflow generation methods see Wu, Annu. Rev. Fluid Mech. 2017. 49:23–49 and Stevens and Meneveau, Annu. Rev. Fluid Mech. 2017. 49:311-39 for applications of such methods to wind farm applications.

b. Figure 1: The description of figure 1, and in particular the description of the strangely oriented circle, took me quite some time to understand. The description of this figure should be improved.

c. Can the method only be applied when the actual computational domain is cylindrical?

d. Figure 3: It is unclear what the green grid cell is. Please clarify.

e. Figure 5: The caption states the velocity field at 40 meters is given. What is meant? The velocity field at 40 meters from the ground level indicated in panel 5a?

f. Figure 6: Can it be indicated in the figure which calculations are performed simultaneously (primal and adjoint solver?) or are the different blocks in the figure performed sequently? If so, in what order.

g1. The section starting with "As it was explained in" on page 13 is rather vague. It is unclear to me what the smoothing function does exactly. I think the authors should explain this in more detail, so readers would be able to implement this part of the solution method by themselves. g2. The fitted profile in figure 3b is neither a logarithmic nor a power law. How exactly is the 1D inflow generating domain adjusted to achieve this? How strong can the deviation from the logarithmic/power law be before the code becomes unstable? How would one run the simulation in such a case (one may end up in an infinite loop in the diagram outlined in figure 6).

h. Does the computational time required by the adjoint solved depend on the initial wind direction and speed that is selected?

i1. Figure 8: It is not entirely clear to me what exactly is meant by the reference profile in figure 8a. i2. Page 14 just below figure 8 states: "Indeed, the output of the optimizer could be the exact reference profile if the convergence criterion was stricter." I am not sure what is meant here. When I look at figure 8b it seems that the calibration is performed on the velocity profile over the hill and this seems to match quite closely. When the velocity profile is calibrated at a specific location it could mean that any deviations with respect to measurements, caused by the used simulation method, would accumulate at another location in the domain (for example at the inflow). Is something like this happening? It would be good to discuss how the solution reacts to this.

j. At the end of the manuscript the authors mention various extensions of the method. I am unsure whether various of these effects could be represented using this methods. Due to the use of the 1D domain to generate the vertical profile means that there is no information on the three-dimensional structure of the flow. It is known that for various properties of the atmospheric boundary layer capturing this three dimensional structure is crucial. I do not see who that can easily be incorporated in this method. Can the authors discuss in more detail what the effect of missing some of the three dimensional flow statistics, spatial flow correlations in the inflow are lost, is?

Minor Page 2: page 3 von Karman ==> von Kármán Page 9: clarity, the Eq. 38 ==> remove "the" Page 11: the frozen turbulent ==> frozen turbulence hypothesis Page 13: "In this way the turbulence model parameters are also gradually updated toward the end the of optimization when the inlet boundary velocities have reached their optimum value." ==> sentence does to flow well, please correct.

---

## Referee Comment (RC2) · Anonymous Referee #2 · 6 Mar 2019

This manuscript discusses a method of determining the inflow conditions needed to match measured wind velocities for wind resource assessments or wind farm design. The proposed adjoint-based method is a reasonable and efficient approach to this problem and would be useful to readers of the journal. Significant reworking of the manuscript, however, would greatly improve the clarity and usefulness of the paper.

1) In general, I found the notation confusing and difficult to understand. In some cases I was unable to follow the derivations because of the confusing notation. Below are a few specific suggestions to clarify or reword the notation:

a) It would be clearer if vectors and tensors were clearly identified in the text. Using

Gibbs notation, either denote vectors as boldface italic and tensors in boldface or use vector symbols above. Alternatively, use index notation. For example, it's confusing to differentiate between the vector $V$ and (scalar?) $V(z)$.

b) What is meant by the subscripts $x, y, z$ in equation 6?

f) Equation 8: Why not use an equality instead of the right arrow?

c) $V(z)$ is never defined. I assume it is the magnitude of the planar-averaged velocity vector.

d) Equation 12: I'm not sure what is meant by $(U, q)R$. I would guess that $U$ and $R$ are vectors and that term can be expanded as $U_x R_1 + U_y R_2 + U_z R_3 + q R_4$.

e) In equations (14)-(16) What do $\delta V$ $\delta p$ mean? What are $J_\Gamma$ and $J_\Omega$? Without knowing what this means, I was unable to follow the adjoint equation derivation.

2) I would significantly rework the structure of the paper to better integrate related ideas. As written, I'm not sure that someone could replicate this algorithm from the details of the paper.

(a) I would combine Section 3-5. Section 3 and 4 are closely related since the adjoint equations are needed to calculate the gradient. Section 5 is part of the adjoint equation derivation and should be included in Section 4. As it stands, it's hard to follow which sections are part of the adjoint-based optimization method description.

(b) Pg. 12 line 16 through pg. 13, Figure 6, and parts of pg. 9 lines 14-19 are related to the flow solver. I would put these details at the top of Section 2: "Flow Model" or with the details of the adjoint equations and gradient-based solver. It would be particularly helpful to have the k-$\varepsilon$ turbulence model mentioned in Section 2.

The suggestion that the structure of the paper could be improved was common among the reviewers. The new manuscript is restructured as follows: The gradient-based optimization and the theory of adjoint method are briefly presented in Section 2. The derivation of the adjoint equations and its BCs from the primal flow model is explained in Section 3. Finally, the numerical results and the conclusions are presented in Sections 4 and 5.

3) This method appears to be related to existing methods in meteorological applications (3DVar) or in existing wind energy papers (see the 4DVar implementation in Bauweraerts and Meyers, BLM, 2019). This is touched on in the introduction, but a more

explicit discussion of how your method is related to existing approaches would be helpful.

The existing 4DVar implementation by **?** is one of the recent studies related to this subject. Instead of calibration of the inflow boundary, they optimize the whole initial field of the domain. They have developed an adjoint solver for a LES-based primal flow model and calibrate the initial flow field with the LIDAR measurements data from the whole of the domain. This is now mentioned in the introduction.

4) Is Section 2.1 used directly in the paper? My understanding is that these inflow conditions are calculated from your algorithm. This section may not really be necessary if that is the case. Also, how are equations (3) and (4) used simultaneously? What is $z_{ref}$? How do you get $n$?

5) pg. 4, line 27: What do you mean by "error-prone"? Finite differencing is simply too expensive to really be used in a gradient-based algorithm. I think that is sufficient justification for using an adjoint-based derivation.

To avoid confusion, and based on some other comments, this has been deleted. The intention was to mention some of the disadvantages of the FD method. As said, the FD is too expensive for CFD gradient-based optimization. However, its usage should not completely be ruled out. For instance, there have been some studies on the application of FD to some selective terms in the discrete adjoint differentiation of CFD solvers (see for example *An aerodynamic design optimization framework using a discrete adjoint approach with OpenFOAM* by He et al., 2018).

6) Top of p. 5: It's not really multiplication, but the inner product of the state equations and the adjoint variables.

7) p. 5, line 22: What is the effect of neglecting changes in eddy viscosity? Shouldn't changing the inflow conditions change the eddy viscosity of the simulation. How is assuming "frozen-turbulence" relevant to the RANS model?

8) Beginning of Section 4: Adjoint methods are generic and applicable to many problems. I don't think it's necessary to point out the specific differences between your application and the Othmer's application.

9) I'm not sure that Section 6 is really necessary. Showing that the gradient-based solver can find a solution is sufficient to demonstrate that the method works.

10) It would be nice to show the application of the optimized inflow boundary condition for evaluating a specific site's wind resources or designing a wind farm. This is the real application and importance of this work, so I would make this a bigger point with an example.

---

## Referee Comment (RC3) · Anonymous Referee #3 · 11 Mar 2019

The authors present their research on calibration of inlet boundary conditions for atmospheric CFD using adjoint gradient-based optimization techniques. The methodology is certainly useful and of interest to the general community. However, I believe the quality of the paper can be increased significantly if the authors address the comments below:

1. I believe the paper would benefit from the addition of some more references in selected places, i.e.:

a) Section 1 - lines 20 - 25, where the authors mention that their adjoint approach has not been applied before in the framework of wind resource assessment. It would be

nice to add some references to recent works where adjoint optimization is used in the context of wind energy. e.g.

blade shape optimization: Dhert, Ashuri, Martins, Wind Energy 2017

wind-farm control: Goit & Meyers, J Fluid Mech 2016; Munters & Meyers, Phil Trans Roy Soc A, 2017; Vali et al., Control Engineering Practice 2019

wind-farm layout optimization: King et al., Wind Energy Science 2017

b) Section 3 - 3.1.1, where the adjoint method is introduced through the formal Lagrange method. A reference to e.g. Hinze, Michael, et al. Optimization with PDE constraints. Vol. 23. Springer Science & Business Media, 2008 seems appropriate.

Also, the explanation of the necesseity and philosphy of the adjoint method is quite poor. The authors could improve this by explicitly mentioning that eq. 11 is expensive since the term d\psi/d\alpha requires a PDE simulation for every dimension in alpha, and showing explicitly that this term drops out in the adjoint method.

I advise the authors to either expand on their explanation of the adjoint methodology, or to refer in to references where it is explained in detail.

2. page 2, lines 3 - 4: The authors first mention the disadvantage of the FD method as being error-prone. In context of the current manuscript, this is misleading in my opinion.

The loss of accuracy due to finite-precision arithmetic can be circumvented by using a complex-step finite differentiation. Also, since the authors use the continuous adjoint method without a grid convergence study, the computed adjoint gradients could certainly be less accurate than a finite-difference approximation.

I feel the authors should remove this claim, or at least put more emphasis on the fact that FD computational cost scales with the input dimensionality, whereas this is not the case for the adjoint method.

3. page 3, line 10: ABL flow simulations –> ABL RANS simulations. Please add the term RANS here, to avoid confusion with the generation of inflow conditions for turbulence-resolving simulations (DNS/LES), which is a whole research field on its own.

4. page 11, line 1: "For gradient evaluation, the 1D velocity profile inflow is rotated by $30°$". This seems like a very large step for a finite-difference gradient approximation. Why did the authors not take a much smaller rotation, e.g. of $1°$? Intuitively, $1°$ still seems large enough to avoid influence of round-off errors.

5. page 11, around line 10: The statement: "most importantly, their signs show that they can be used for the purpose of gradient-based calibration" is misleading. The authors seem to state that, in laymans terms, having a gradient that point approximately in the right direction is sufficient for optimization. However, this claim should be nuanced. Gradient inaccuracies can severely impact the performance, stability and convergence of a given optimization algorithm. For instance, in quasi-Newton methods this could lead to instabilities because of poor Hessian approximations, and in CG methods this could lead to non-conjugate search directions in successive iterations.

Furthermore, related to comment nr. 2 and comment nr. 4, I feel the authors should be careful in attributing discrepancies between adjoint and FD gradients to inaccuracies of the FD gradient. Intuitively, I would expect the frozen turbulence assumption and the grid resolution (combined with continuous adjoint approach) to be the main reasons for discrepancies.

6. page 12, line 18: The authors mention some facts about computational cost of their simulations. These facts can be made more illustrative by also explicitly mentioning the walltime of a primal flow run, and explaining why the run-time of the adjoint solver is 60% of the primal (e.g. because the adjoint equations are linear)?

7. (suggestion) page 13, line 5: "The optimizer may ask for a cost function evaluation with a new inflow boundary which is highly unrealistic for an ABL domain. ... the curve fit capability is used to smooth and fit the new inflow to a boundary which has

a log/power law characteristic." The authors manually post-process the new iterate of inflow conditions during the optimization process. Although I agree that it is undesirable to run RANS with unrealistic inflows during optimization, this manual postprocessing will can have a significant detrimental impact on the convergence of the optimization.

Since this post-processing imposes a log/power law profile, it seems more natural to directly use the parameters for such log/power profiles as decision variables, in contrast to optimizing the individual inflow velocities at every height. This would directly inform the optimizer of the desired log/power law profile, and could improve convergence a lot.

8. (suggestion) page 13, line 18: "This can be explained by the fact that in early iterations the derivative of cost function wrt WD is much bigger than wrt inflow". This spike might be avoided and convergence could possibly be improved by using quasi-Newton optimization methods (e.g. BFGS), this is a suggestion for future applications of the methodology.

9. Figure 2 - avoid rainbow colormaps, which can be misleading and are unintelligible in black & white. Use a standard perceptually uniform colormap such as parula (Matlab) or viridis (matplotlib). Same for Figure 5

10. Typographically & gramatically the paper needs to be proofread in detail. For instance,

- eq. 7: is missing a parenthesis ( before V_M_i

- eq. 15: is missing a parenthesis ( before \delta V

- page 5, line 15: as followings -> as following

- page 11, line 9: higher that -> higher than

- page 11, line 11: because at it was shown in -> because it was shown in

... and quite some more.

---

## Referee Comment (RC4) · Anonymous Referee #4 · 14 Mar 2019

The manuscript employs gradient- and adjoint-based method for calibration of inlet velocity profiles for ABL simulation. To that end, cost function for the optimization problem is defined such that the simulated velocity and the target velocity at the observation point in the simulation domain show maximum agreement for the optimized inflow profile. As a test case, the authors verify their method by applying it to a domain located close to Kassel in Germany. Accurate prediction and estimation of inflow condition is of great interest for wind farm simulation as well as for ABL research in general. Like many other methods (e.g. precursor simulation with re-scaled velocity field), gradient-based optimization of inflow profile can also be one of the approaches for inflow generation. Also, if this method works perfectly, it will have advantage over other methods in terms

of accuracy, as it does not make any big assumption. However, adjoitn-based methods have their own limitation; for instance, they are not very stable for realistic problems. The author may discuss this and other issues that they may have experienced during their simulation in the manuscript.

Although the issue addressed in the manuscript is very interesting, the manuscript need a lot of technical as well as editorial modification before it is ready for publication. First of all, it seems you do not have convective terms in the adjoint equation. If you have a reason for removing the convective terms please discuss that, else if it is a mistake then I suggest that you redo all the simulations. Next, as I have also put in the specific comments, LESs have become more common for ABL simulations and generation of time-dependent inflow data is more of the issue. Therefore, my concern is why did you choose RANS in this study instead of LES? On the editorial side, you have too many small sections and subsections, and information are spread all over those sections (in particular in section 2, 3, 4 and 5). So, I suggest that you further organize the manuscript.

**Specific comments:**

1. The purpose of section 2 is not clear. Do you want to give general description of inflow boundary condition and forest effect, or are these the techniques you will use in your study? If it is the former then they should go to Introduction, if it is the later, then you should explain them together with the discussions in Section 3 and Section 4.

2. You have Section 3, subsection 3.1 and sub subsection 3.1.1, but no following subsection (e.g. 3.2 etc) or following sub subsection (e.g. 3.1.2 etc). So, I suggest that you put all the contents in this section under a single section heading without any subsection. But my main concern for this section is once again, it is not clear whether you are trying to explain a general method for gradient
evaluation in optimization or is it for your specific problem? The section lacks explanation that may be necessary for some one not familiar with Gradient-based optimization. Therefore, some more discussion will be required. For example, what is the role of design variable $\alpha$, what will your algorithm do to optimize it and why can you write Eq. (11). Furthermore, last sentence in section 3.1 (line 28 and 29) will not come as obvious to many readers.

3. Section 4, 1st Paragraph: The purpose of this paragraph is not clear. Summarizing the differences between your work and that of earlier work is not really necessary. Because the two works deal with different optimization problem, all three differences stated in the manuscript are obvious.

4. The forest model, Eq. (18) should be a part of original Lagrangian and should also appear in Eq. (15) and (16).

5. Why do you not have convective (and cross-convective) terms in Eq. (19)? I still see them in Eq. (16). If it is an error, then please correct it. If you have a proper reason why they can be neglected, you must explain that.

6. It is not clear what $\omega_j$ is.

7. It has become more common (at least in academic researches) to use large-eddy simulations (LES) for ABL and wind farm simulations. But authors preferred to use RANS in their work. Can you please discuss why you chose RANS? Was it because RANS is cheaper or was it because it is easy to implement adjoint equations for RANS problem? In reality both inflow profiles and measured velocity at the measurement points $V_M$ will vary with time. Furthermore, dynamic behavior of flow field as well and wind farm are receiving more interest in wind energy community (e.g. farm level controller). So, it seems LES would have been preferred simulation method.
8. Pg. 12, Line 14–15: It is not clear why you used velocity profile from a reference simulation instead of the velocity measured by the met-mast. Also, if your reference simulation and actual simulation were performed in the similar condition, then you will obviously get good optimization result. Please provide further information regarding this issue.

9. Pg. 12, Line 16–18: You need to provide more information about conjugate-gradient algorithm.

10. Pg. 12, Line 19: Are you sure that the run-time of adjoint equations is 60% of the primal equations? For most work that I am aware of and from my personal experience, adjoint equations always took longer time to simulate.

11. Pg. 13, Line 4–10: You are fitting the inlet boundary condition from the optimization to a logarithmic or a power law. This may not be a good idea, if you want to exploit the full potential of your optimization scheme. Therefore, instead why do not you add some sort of constrain to your system or add appropriate penalty term to the cost functional?

12. I do not think you have presented sufficient result to consider this manuscript as a technical paper. You only have figure 8 as the results for one simulation case. Please define and perform optimization for more simulation cases. Also, you need to provide further discussion of your result.

**Editorial suggestions:**

1. As I have mentioned earlier, you have too many sections and subsections and discussions are mixed up in sections 2–5. For example, the discussion about optimization package will be more suitable in section 4. Section 5 and Section 4 can be combined.

2. Pg. 1, Line 16: ... direction and etc. → ... direction etc.

3. Pg. 2, Line 2: cost function evaluations → function evaluations
   Just for consistency

4. Pg. 2, Line 12: ... line-by-line differentiating of ... → ... line-by-line differentiation of ...

5. Eq. (3): I do not think capital $V$ is a commonly used notation for velocity. In major books of fluid mechanics as well as in papers, I usually see $u(z)$.

6. You may want to add schematic for cylindrical domain discussed in section 6.1. First of all definition of x, y, z is not clear. Next, cross sectional (x-z) view in figure 3 is not clear?

7. Pg. 10: If wind blows from east to west then wind direction is $90°$ and not $270°$.

8. Pg. 11, Line 9: higher that w.r.t. → higher than w.r.t.

9. Pg. 13, Line 13: the end the of optimization → the end of the optimization

---

## Author Response (AR1)

**The Authors response to the reviewers' comments**

The authors would like to thank the referees for carefully reading our manuscript and for giving such detailed comments which substantially helped to improve the quality of the paper. In the revised version of the manuscript, we have tried to address all the points that were raised. The points that relate to the text's language and grammar are not discussed here. In the following, the comments will be discussed one by one.

**- Anonymous Referee #1**

a) Page 2 states "Mainly two different methods are used to set the inflow boundary conditions for ABL flow simulations". There are various other methods that are used within the community such as using white noise, Mann spectrum, (concurrent) precursor methods, etc. For a detailed discussion of turbulence inflow generation methods see Wu, Annu. Rev. Fluid Mech. 2017. 49:23–49 and Stevens and Meneveau, Annu. Rev. Fluid Mech. 2017. 49:311-39 for applications of such methods to wind farm applications.

It was not the intention of the authors to say that the mentioned methods are the only available solutions to generate an inflow boundary condition for CFD solvers. However, it was not probably clear in the text that we are discussing in the context of the RANS steady simulations. The inflow boundary generation, which is now in Section 3.1.1, refers to the suggested scientific works and does not include the explanation of the analytical function approaches.

- b) Figure 1: The description of figure 1, and in particular the description of the strangely oriented circle, took me quite some time to understand. The description of this figure should be improved.
   Both the description in the text and the figure itself are improved.
- c) Can the method only be applied when the actual computational domain is cylindrical?

The method can be applied to other computational domain which does not have cylindrical shapes. It is now mentioned in the text. Moreover, the added Ishihara test case proves this too.

d) Figure 3: It is unclear what the green grid cell is. Please clarify.

The figure is changed (see figure 2), and the forest part is clear now.

e) Figure 5: The caption states the velocity field at 40 meters is given. What is meant? The velocity field at 40 meters from the ground level indicated in panel 5a?

It is now the figure 9. It shows the velocity field on a plane which has a 40 [m] perpendicular distance from the terrain's surface points (basically a slice of the domain that follows the terrain).

f) Figure 6: Can it be indicated in the figure which calculations are performed simultaneously (primal and adjoint solver?) or are the different blocks in the figure performed sequentially? If so, in what order.

The flow chart is updated, and it is now mentioned that all steps are performed sequentially. The optimizer decides if the gradient or the cost function value is needed. Then the corresponding path and steps will be followed.

10

15

20

30

g) 1. The section starting with "As it was explained in" on page 13 is rather vague. It is unclear to me what the smoothing function does exactly. I think the authors should explain this in more detail, so readers would be able to implement this part of the solution method by themselves.

2. The fitted profile in figure 3b is neither a logarithmic nor a power law. How exactly is the 1D inflow generating domain adjusted to achieve this? How strong can the deviation from the logarithmic/power law be before the code becomes unstable? How would one run the simulation in such a case (one may end up in an infinite loop in the diagram outlined in figure 6).

The profile in figure 3b is the profile in the middle of the forest and above it. The velocity for the heights above the forest is used for gradient validation, not the inflow calibration. The logarithmic or power-law function discussion is about the inflow boundary, not the profile inside of the domain.

The smoothing is now explained in Section 4.2. The output of the optimizer is first filtered to avoid sharp spikes. Then it is checked if it is close enough to a logarithmic or power-law function. If either of these functions is fitted and its coefficient of determination is above 0.96 the smoothed inflow from optimizer (not the fitted profile!), is accepted for the CFD solver. Otherwise, the optimization takes the last fitted profile and asks for a new gradient evaluation. In this way, the inflow boundary is not necessarily a logarithmic or power-law profile, and, moreover, it is not so unrealistic to be problematic for the solver. As mentioned in the reviewer's comments and in the new manuscript, alternatively, constraints or penalization term can be added to the objective function. This will be explored in future works when for instance the inflow turbulence properties are also considered as design parameters.

h) Does the computational time required by the adjoint solved depend on the initial wind direction and speed that is selected?

Of course, the adjoint solver inherits some of the properties of the original solver. The inflow wind speed and direction affect the computational time of the primal solver. Subsequently, the run time of the adjoint would be some how dependent on these parameters. In other word, it is case dependent.

i) 1. Figure 8: It is not entirely clear to me what exactly is meant by the reference profile in figure 8a.

It is the target inflow boundary. The legend of the plot is updated (see figure 11).

2. Page 14 just below figure 8 states: "Indeed, the output of the optimizer could be the exact reference profile if the convergence criterion was stricter." I am not sure what is meant here. When I look at figure 8b it seems that the calibration is performed on the velocity profile over the hill and this seems to match quite closely. When the velocity profile is calibrated at a specific location it could mean that any deviations with respect to measurements, caused by the used simulation method, would accumulate at another location in the domain (for example at the inflow). Is something like this happening? It would be good to discuss how the solution reacts to this.

Although a tighter error criterion would improve the accuracy of the optimized profile, it also increases the calibration iterations and subsequently the number of CFD solver calls. The sensitivity of wind speed at a certain

10

5

15

point in the domain to a very small change in the inflow boundary is dependent on many parameters such as terrain complexity, wind direction, CFD model, etc., and cannot be easily generalized.

j) At the end of the manuscript the authors mention various extensions of the method. I am unsure whether various of these effects could be represented using this methods. Due to the use of the 1D domain to generate the vertical profile means that there is no information on the three-dimensional structure of the flow. It is known that for various properties of the atmospheric boundary layer capturing this three dimensional structure is crucial. I do not see how that can easily be incorporated in this method. Can the authors discuss in more detail what the effect of missing some of the three dimensional flow statistics, spatial flow correlations in the inflow are lost, is?

In general, adding the differentiation of the turbulence model with thermal stratification and Coriolis force to the current adjoint model would definitely improve the accuracy of the calibration. Moreover, the usage of the 1D inflow does not reflect the 3D structure of the flow. In order to overcome this limitation, this method could be extended to optimize the inlet BC as a spatially 2D field with three velocity components. However, a more sophisticated smoothing or penalization is needed to avoid having an unrealistic inlet field.

**- Anonymous Referee #2**

- 1. In general, I found the notation confusing and difficult to understand. In some cases I was unable to follow the derivations because of the confusing notation. Below are a few specific suggestions to clarify or reword the notation:
  - a) It would be clearer if vectors and tensors were clearly identified in the text. Using Gibbs notation, either denote vectors as boldface italic and tensors in boldface or use vector symbols above. Alternatively, use index notation. For example, it's confusing to differentiate between the vector V and (scalar?) V (z).
  - b) What is meant by the subscripts x, y, z in equation 6?
  - c) V(z) is never defined. I assume it is the magnitude of the planar-averaged velocity vector.
  - d) Equation 12: I'm not sure what is meant by (U, q)R. I would guess that U and R are vectors and that term can be expanded as UxR1 + UyR2 + UzR3 + qR4.
  - e) In equations (14)-(16) What do  $\delta V$  and  $\delta p$  mean? What are  $J_{\Gamma}$  and  $J_{\Omega}$ ? Without knowing what this means, I was unable to follow the adjoint equation derivation.
  - f) Equation 8: Why not use an equality instead of the right arrow? (equality is now used in Eq. (1))

The suggestions are considered in the new manuscript:

- The italic bold letter is a vector (e.g.  $U, \overline{U}$ ) and the normal bold letter stands for a tensor (**D**).
- The terms with  $\delta$  removed and all the equations are represented and explained by derivative symbol,  $\partial$ .
- V(z) is not anymore in the text.

-  $\Gamma$  and  $\Omega$  represent the boundary and the volume of the computational domain respectively. As a result,  $J_{\Gamma}$  and  $J_{\Omega}$  are the part cost function which is dependent on flow state values on the boundary and the interior of the domain. This has been clarified in the manuscript.

10

15

5

25

20

- 2. I would significantly rework the structure of the paper to better integrate related ideas. As written, I'm not sure that someone could replicate this algorithm from the details of the paper.
  - a) I would combine Section 3-5. Section 3 and 4 are closely related since the adjoint equations are needed to calculate the gradient. Section 5 is part of the adjoint equation derivation and should be included in Section 4. As it stands, it's hard to follow which sections are part of the adjoint-based optimization method description.
  - b) Pg. 12 line 16 through pg. 13, Figure 6, and parts of pg. 9 lines 14-19 are related to the flow solver. I would put these details at the top of Section 2: "Flow Model" or with the details of the adjoint equations and gradient-based solver. It would be particularly helpful to have the k- $\varepsilon$  turbulence model mentioned in Section 2.

The suggestion that the structure of the paper could be improved was common among the reviewers. The new manuscript is restructured as follows: The gradient-based optimization and the theory of adjoint method are briefly presented in Section 2. The derivation of the adjoint equations and its BCs from the primal flow model is explained in Section 3. Finally, the numerical results and the conclusions are presented in Sections 4 and 5.

3. This method appears to be related to existing methods in meteorological applications (3DVar) or in existing wind energy papers (see the 4DVar implementation in Bauweraerts and Meyers, BLM, 2019). This is touched on in the introduction, but a more explicit discussion of how your method is related to existing approaches would be helpful.

The existing 4DVar implementation by Bauweraerts and Meyers is one of the recent studies related to this subject. Instead of calibration of the inflow boundary, they optimize the whole initial field of the domain. They have developed an adjoint solver for a LES-based primal flow model and calibrate the initial flow field with the LIDAR measurements data from the whole of the domain. This is now mentioned in the introduction.

4. Is Section 2.1 used directly in the paper? My understanding is that these inflow conditions are calculated from your algorithm. This section may not really be necessary if that is the case. Also, how are equations (3) and (4) used simultaneously? What is *zref*? How do you get *n*?

**This has been clarified in Section 4.2.**

5. pg. 4, line 27: What do you mean by "error-prone"? Finite difference is simply too expensive to really be used in a gradient-based algorithm. I think that is sufficient justification for using an adjoint-based derivation.

To avoid confusion, and based on some other comments, this has been deleted. The intention was to mention some of the disadvantages of the FD method. As said, the FD is too expensive for CFD gradient-based optimization. However, its usage should not completely be ruled out. For instance, there have been some studies on the application of FD to some selective terms in the discrete adjoint differentiation of CFD solvers (see for example *An aerodynamic design optimization framework using a discrete adjoint approach with OpenFOAM* by He et al., 2018).

Top of p. 5: It's not really multiplication, but the inner product of the state equations and the adjoint variables.
 Corrected.

10

5

25

30

- 7. p. 5, line 22: What is the effect of neglecting changes in eddy viscosity? Shouldn't changing the inflow conditions change the eddy viscosity of the simulation. How is assuming "frozen-turbulence" relevant to the RANS model? There is no doubt that the turbulence contribution to the adjoint equations cannot simply be ignored. In some applications, frozen-turbulence can produce the wrong sign for the local sensitivity (Zymaris et al., 2009; Papoutsis-Kiachagias et al., 2015). In the context of inflow calibration, the authors believe that the differentiation of the turbulence model is better to be included when the inflow turbulence properties are also design parameters, which
- 8. Beginning of Section 4: Adjoint methods are generic and applicable to many problems. I don't think it's necessary to point out the specific differences between your application and the Othmer's application.
- This paragraph is removed.

can be explored in future works.

9. I'm not sure that Section 6 is really necessary. Showing that the gradient-based solver can find a solution is sufficient to demonstrate that the method works.

As one of the reviewers has pointed out, the accuracy of the gradient computation is an important element of any gradient-based optimization. As a common practice in scientific studies, the validation of a newly implemented adjoint solver is presented.

10. It would be nice to show the application of the optimized inflow boundary condition for evaluating a specific site's wind resources or designing a wind farm. This is the real application and importance of this work, so I would make this a bigger point with an example.

This point has been raised by other reviewers. Due to time limits and available sites in the project, this was not considered for this study. For sure this will be tried in the future.

**- Anonymous Referee #3**

- 1. I believe the paper would benefit from the addition of some more references in selected places, i.e.:
  - a) Section 1 lines 20-25, where the authors mention that their adjoint approach has not been applied before in the framework of wind resource assessment. It would be nice to add some references to recent works where adjoint optimization is used in the context of wind energy. e.g. blade shape optimization: Dhert, Ashuri, Martins, Wind Energy 2017 wind-farm control: Goit & Meyers, J Fluid Mech 2016; Munters & Meyers, Phil Trans Roy Soc A, 2017; Vali et al., Control Engineering Practice 2019 wind-farm layout optimization: King et al., Wind Energy Science 2017
  - b) Section 3 3.1.1, where the adjoint method is introduced through the formal Lagrange method. A reference to e.g. Hinze, Michael, et al. Optimization with PDE constraints. Vol. 23. Springer Science & Business Media, 2008 seems appropriate. Also, the explanation of the necessity and philosophy of the adjoint method is quite poor. The authors could improve this by explicitly mentioning that eq. 11 is expensive since the term  $\frac{d\psi}{d\alpha}$

10

5

20

15

30

requires a PDE simulation for every dimension in alpha, and showing explicitly that this term drops out in the adjoint method. I advise the authors to either expand on their explanation of the adjoint methodology, or to refer in to references where it is explained in detail.

The recommended references are added to the corresponding sections. The adjoint method explanation is improved.

5

10

15

20

25

30

2. page 2, lines 3 - 4: The authors first mention the disadvantage of the FD method as being error-prone. In context of the current manuscript, this is misleading in my opinion. The loss of accuracy due to finite-precision arithmetic can be circumvented by using a complex-step finite differentiation. Also, since the authors use the continuous adjoint method without a grid convergence study, the computed adjoint gradients could certainly be less accurate than a finite-difference approximation. I feel the authors should remove this claim, or at least put more emphasis on the fact that FD computational cost scales with the input dimensionality, whereas this is not the case for the adjoint method.

It has been mentioned in the manuscript that the complex-step method would circumvent the difficulties of FD. However, it should also be noted that the implementation of a complex-step method on a large code which is not written with complex variables (e.g. OpenFOAM) is not straight forward too.

In the manuscript, the emphasis is put now on the advantages of the adjoint method in terms of the computational cost for a large number of design parameters.

3. page 3, line 10: ABL flow simulations -> ABL RANS simulations. Please add the term RANS here, to avoid confusion with the generation of inflow conditions for turbulence-resolving simulations (DNS/LES), which is a whole research field on its own.

Corrected. It is now in section 3.1.1.

- 4. page 11, line 1: "For gradient evaluation, the 1D velocity profile inflow is rotated by 30°". This seems like a very large step for a finite-difference gradient approximation. Why did the authors not take a much smaller rotation, e.g. of 1°? Intuitively, 1° still seems large enough to avoid influence of round-off errors.
- Please be aware that, the 30° is not the step-size. First, a simulation is carried out with WD=270°. The velocities, above the forest, are taken as pseudo measurements. Then, using these measurements, the adjoint gradient is computed for the simulation with WD=240°. The finite difference gradient is also carried out for WD=240° in which the step-size is  $0.2^{\circ}$ .
  - 5. page 11, around line 10: The statement: "most importantly, their signs show that they can be used for the purpose of gradient-based calibration" is misleading. The authors seem to state that, in laymans terms, having a gradient that point approximately in the right direction is sufficient for optimization. However, this claim should be nuanced. Gradient inaccuracies can severely impact the performance, stability and convergence of a given optimization algorithm. For instance, in quasi-Newton methods this could lead to instabilities because of poor Hessian

approximations, and in CG methods this could lead to non-conjugate search directions in successive iterations. Furthermore, related to comment nr. 2 and comment nr. 4, I feel the authors should be careful in attributing discrepancies between adjoint and FD gradients to inaccuracies of the FD gradient. Intuitively, I would expect the frozen turbulence assumption and the grid resolution (combined with continuous adjoint approach) to be the main reasons for discrepancies.

No doubt the accuracy of the gradient is as important as its sign due to the reasons which are mentioned above. Otherwise, we would not put such a section in the manuscript. However, the words should have been chosen more carefully.

The point regarding the FD error is considered in the manuscript.

- 6. page 12, line 18: The authors mention some facts about computational cost of their simulations. These facts can be made more illustrative by also explicitly mentioning the wall-time of a primal flow run, and explaining why the run-time of the adjoint solver is 60% of the primal (e.g. because the adjoint equations are linear)? The wall-time of the primal run is given in the new manuscript. Please also refer to the following response: - Anonymous Referee #4, question: 10
- 15 7. (suggestion) page 13, line 5: "The optimizer may ask for a cost function evaluation with a new inflow boundary which is highly unrealistic for an ABL domain. ... the curve fit capability is used to smooth and fit the new inflow to a boundary which has a log/power law characteristic." The authors manually post-process the new iterate of inflow conditions during the optimization process. Although I agree that it is undesirable to run RANS with unrealistic inflows during optimization, this manual post-processing will can have a significant detrimental impact 20 on the convergence of the optimization. Since this post-processing imposes a log/power law profile, it seems more natural to directly use the parameters for such log/power profiles as decision variables, in contrast to optimizing the individual inflow velocities at every height. This would directly inform the optimizer of the desired log/power law profile, and could improve convergence a lot.

As it is explained in the manuscript and before in this document, the point of smoothing is to avoid enforcing the boundary to have either a logarithmic or power-law function shape. Otherwise, optimizing the function parameters would be straightforward. As in the Ishihara case, the inflow boundary can be neither a logarithmic nor a powerlaw, but still being able to reproduce the measurements with acceptable tolerance. Of course, the approach can be improved by for example adding corresponding constraints to the objective function.

8. (suggestion) page 13, line 18: "This can be explained by the fact that in early iterations the derivative of cost function wrt WD is much bigger than wrt inflow". This spike might be avoided and convergence could possibly be improved by using quasi- Newton optimization methods (e.g. BFGS), this is a suggestion for future applications of the methodology.

Thank you for the helpful suggestion. We think also this will help.

10

5

25

**- Anonymous Referee #4**

**Specific comments:**

 The purpose of section 2 is not clear. Do you want to give general description of inflow boundary condition and forest effect, or are these the techniques you will use in your study? If it is the former then they should go to Introduction, if it is the later, then you should explain them together with the discussions in Section 3 and Section 4.

This part is now moved to the section 4.2 where the optimization steps is discussed.

2. You have Section 3, subsection 3.1 and sub subsection 3.1.1, but no following subsection (e.g. 3.2 etc) or following subsection (e.g. 3.1.2 etc). So, I suggest that you put all the contents in this section under a single section heading without any subsection. But my main concern for this section is once again, it is not clear whether you are trying to explain a general method for gradient evaluation in optimization or is it for your specific problem? The section lacks explanation that may be necessary for some one not familiar with Gradient-based optimization. Therefore, some more discussion will be required. For example, what is the role of design variable α, what will your algorithm do to optimize it and why can you write Eq. (11). Furthermore, last sentence in section 3.1 (line 28 and 29) will not come as obvious to many readers.

The new restructured manuscript should not have this problem. Moreover, both the gradient-based (section 2) and the CG optimizer (4.2) are explained in the paper.

3. Section 4, 1st Paragraph: The purpose of this paragraph is not clear. Summarizing the differences between your work and that of earlier work is not really necessary. Because the two works deal with different optimization problem, all three differences stated in the manuscript are obvious.

**The paragraph is removed.**

- The forest model, Eq. (18) should be a part of original Lagrangian and should also appear in Eq. (15) and (16). Corrected.
- 5. Why do you not have convective (and cross-convective) terms in Eq. (19)? I still see them in Eq. (16). If it is an error, then please correct it. If you have a proper reason why they can be neglected, you must explain that.

It is not used anymore in the new manuscript (see Eq. (21)) but

 $-2\mathbf{D}(\overline{\boldsymbol{U}})\boldsymbol{U} = -\nabla\overline{\boldsymbol{U}}\cdot\boldsymbol{U} - (\boldsymbol{U}\cdot\nabla)\overline{\boldsymbol{U}}$

6. It is not clear what  $\omega_i$  is.

It is the volume of the cell in which the measurement is located.

10

5

15

20

25

7. It has become more common (at least in academic researches) to use large-eddy simulations (LES) for ABL and wind farm simulations. But authors preferred to use RANS in their work. Can you please discuss why you chose RANS? Was it because RANS is cheaper or was it because it is easy to implement adjoint equations for RANS problem? In reality both inflow profiles and measured velocity at the measurement points VM will vary with time. Furthermore, dynamic behavior of flow field as well and wind farm are receiving more interest in wind energy community (e.g. farm level controller). So, it seems LES would have been preferred simulation method.

As it is mentioned in the new manuscript, there are some studies that use measurements to calibrate the LES-based models (Bauweraerts and Meyers, 2018). However, due to the computational cost, the most common CFD solvers in the wind energy industry is still steady-state RANS-based models. Also, RANS is the model of choice here because of the scope of the current project. We completely agree the LES is gaining momentum due to its superior predictions in many cases compared to RANS.

8. Pg. 12, Line 14–15: It is not clear why you used velocity profile from a reference simulation instead of the velocity measured by the met-mast. Also, if your reference simulation and actual simulation were performed in the similar condition, then you will obviously get good optimization result. Please provide further information regarding this issue.

Indeed, the main idea of this work is using measurements to calibrate the inflow boundary of the solver. For the Ishihara case, the measured wind speed  $U_x$  was used for the optimization process. Since it is a wind tunnel experiment, it is fair to assume the wind direction to be the *x* axis. Although the wind speed at six different heights was available for the Kassel case, the wind direction data was present at only two heights. Therefore, we used the pseudo measurement to prove the validity of the methodology. However, the profile from which the pseudo measurements are selected essentially follows the trend of the real measurements of the domain. Please see the comparison in the figure below, which has not been included in the paper because it may confuse the reader.

9. Pg. 12, Line 16–18: You need to provide more information about conjugate- gradient algorithm.

An elaborate explanation has been added to sec. (4.2).

10. Pg. 12, Line 19: Are you sure that the run-time of adjoint equations is 60% of the primal equations? For most work that I am aware of and from my personal experience, adjoint equations always took longer time to simulate.

Yes, in most of the adjoint simulations that we have carried out with this adjoint solver the run-time is 60-75% of the original flow run-time. A couple of points should be considered: a) The main driver of this adjoint solver is the source term which is the difference between the measurement and simulated results. This source term appears only for a few cells in which the measurements are located. Though, comparing to other adjoint solvers for shape and topology optimization this can be seen as a simpler solver. b) The turbulence is not differentiated; meaning there are no adjoint turbulence equations to be solved.

In the new manuscript, instead of a certain percentage, a range is given.

10

5

15

30

Figure A1. Comparison of the velocity profile over the hill from which the pseudo measurements are chosen and the velocity speeds measured by the met-mast.

11. Pg. 13, Line 4–10: You are fitting the inlet boundary condition from the optimization to a logarithmic or a power law. This may not be a good idea, if you want to exploit the full potential of your optimization scheme. Therefore, instead why do not you add some sort of constrain to your system or add appropriate penalty term to the cost functional?

This point is discussed before in this document. Please refer to the following responses:

- Anonymous Referee #1, question: g
- Anonymous Referee #3, question: 7
- 12. I do not think you have presented sufficient result to consider this manuscript as a technical paper. You only have figure 8 as the results for one simulation case. Please define and perform optimization for more simulation cases. Also, you need to provide further discussion of your result.

The results for the Ishihara et al. test case is also added in the new manuscript. We hope this is sufficient for the scope of the paper. In future work, we will investigate more cases with different complexity. Our main aim here is to provide the general framework and initial case studies.

[revised manuscript text omitted]

Decomposition of parts into interior domain,  $\Omega$ , and its boundaries,  $\Gamma$ , leads to reformulation of Eq. (18) as follows

$$\int_{\Gamma} \left[ \overline{U} \cdot \mathbf{n} + \frac{\partial J_{\Gamma}}{\partial p} \right] \frac{\partial p}{\partial \alpha} d\Gamma + \int_{\Gamma} \left[ \mathbf{n} (\overline{U} \cdot U) + \overline{U} (U \cdot \mathbf{n}) + 2v_{eff} \mathbf{n} \cdot \mathbf{D} (\overline{U}) - \overline{p} \mathbf{n} + \frac{\partial J_{\Gamma}}{\partial U} \right] \frac{\partial U}{\partial \alpha} d\Gamma$$

$$+ \int_{\Gamma} \left[ -2v_{eff} \mathbf{n} \cdot \mathbf{D} (\frac{\partial U}{\partial \alpha}) \cdot \overline{U} \right] d\Gamma + \int_{\Omega} \left[ -\nabla \cdot \overline{U} + \frac{\partial J_{\Omega}}{\partial p} \right] \frac{\partial p}{\partial \alpha} d\Omega$$

$$\quad + \int_{\Omega} \left[ -\nabla \overline{U} \cdot U - (U \cdot \nabla) \overline{U} - \nabla \cdot (2v_{eff} \mathbf{D} (\overline{U})) + \frac{1}{2} C_D A |U| + \nabla \overline{p} + \frac{\partial J_{\Omega}}{\partial U} \right] \frac{\partial U}{\partial \alpha} d\Omega = 0 \tag{19}$$

 $J_{\Gamma}$  and  $J_{\Omega}$  stand, respectively, for the part of the cost function which is dependent on the flow state values on boundary and volume of the domain. Due to the definition of the cost function Eq. (13), its direct variation comes only from the interior domain. Moreover, it does not have any derivative of the pressure field. The corresponding terms are zeroed out in Eq. (19). The only derivative of the cost function is w.r.t the inflow and velocity in the interior of the domain, and at the locations where the measurements are available. From the latter we have

$$rac{\partial J_{\Omega}}{\partial U} = -2(U_{M_i} - U_{S_i}) \quad i = 1, 2, \dots$$

Using Eqs. (19 and 20) the adjoint equations can be derived as

20

$$-\nabla \overline{U} \cdot U - (U \cdot \nabla) \overline{U} = -\nabla \overline{p} + \nabla \cdot (2\nu_{eff} \mathbf{D}(\overline{U})) + (\frac{2}{\omega_i}) (U_{M_i} - U_{S_i}) - \frac{1}{2} C_D A |U| \overline{U}$$
(21)

(20)

$$\nabla \cdot \overline{U} = 0 \tag{22}$$

where  $\omega_i$  is the volume of the cell in which the measurement is located.

**3.2.1 Boundary Conditions**

The boundary integrals of Eq. (19) can be mathematically re-formulated and reduced to

$$\int_{\Gamma} \left[ \overline{U} \cdot \mathbf{n} \right] \frac{\partial p}{\partial \alpha} \, d\Gamma = 0 \tag{23}$$

5
$$\int_{\Gamma} \left[ \boldsymbol{n}(\overline{\boldsymbol{U}} \cdot \boldsymbol{U}) + \boldsymbol{v}_{eff}(\boldsymbol{n} \cdot \nabla)\overline{\boldsymbol{U}} - \overline{p}\boldsymbol{n} \right] \cdot \frac{\partial \boldsymbol{U}}{\partial \alpha} \, d\Gamma - \int_{\Gamma} \left[ \boldsymbol{v}_{eff}(\boldsymbol{n} \cdot \nabla) \frac{\partial \boldsymbol{U}}{\partial \alpha} \cdot \overline{\boldsymbol{U}} \right] \, d\Gamma = 0$$
(24)

where n is the unit normal vector from the boundary faces. Except for the inlet, which is the design space, the adjoint BCs should be chosen such that the above equations are held.

Generally, for an ABL CFD domain no-slip wall (zero fixed velocity) and zero pressure gradient conditions are imposed on the ground. For a wall type of boundary in which  $\frac{\partial U}{\partial \alpha}$  is zero the first integral of Eq. (24) is cancelled. Then, the only way to satisfy the following conditions

$$\overline{U} \cdot \boldsymbol{n} = 0 \tag{25}$$

$$(\boldsymbol{n}\cdot\nabla)\frac{\partial \boldsymbol{U}}{\partial\boldsymbol{\alpha}}\cdot\overline{\boldsymbol{U}}=0$$
(26)

is to apply a no slip ( $\overline{U} = 0$ ) condition on the ground. No BC can be derived on the ground for the adjoint pressure but consistent with the primal a zero gradient condition is applied.

For the top and outlet boundaries of the domain a zero gradient velocity  $((\mathbf{n} \cdot \nabla) \frac{\partial U}{\partial \alpha} = 0)$  and zero fixed pressure (p = 0) are the common conditions for the primal system. These conditions fulfil Eq. (23) and cancel the second integral of Eq. (24). The only term that remains is the first term of Eq. (24) which needs to be zeroed out,

$$\left[\boldsymbol{n}(\overline{\boldsymbol{U}}\cdot\boldsymbol{\boldsymbol{U}})+\boldsymbol{v}_{eff}(\boldsymbol{n}\cdot\nabla)\overline{\boldsymbol{U}}-\overline{p}\boldsymbol{n}\right]\cdot\frac{\partial\boldsymbol{\boldsymbol{U}}}{\partial\boldsymbol{\alpha}}=0$$
(27)

After decomposition into tangent and normal components it can be shown that the relations below should hold

20
$$\overline{p} = \overline{U} \cdot U + U_n U_n + v_{eff} (\mathbf{n} \cdot \nabla) U_n$$
 (28)

$$0 = U_n U_t + v_{eff} (\boldsymbol{n} \cdot \nabla) U_t \tag{29}$$

where subscripts n and t represent the normal and in-plane components respectively. The adjoint BCs can be summarized as

ground (wall):
$$\overline{U} = 0$$
  $\mathbf{n} \cdot \nabla q = 0$  (30)

top/outlet:
$$q = \overline{U} \cdot U + U_n U_n + v_{eff} (n \cdot \nabla) U_n$$
  $U_t = 0$  (31)

25 It is worth mentioning that the last term of the adjoint pressure which includes the kinematic viscosity, in implementation is often neglected (Nilsson et al., 2014). Moreover, the adjoint variables at the inlet should not be chosen to zero out the inlet

velocity perturbations because the design variables are the inlet velocities. Instead, the zero gradient condition is imposed on the inlet for both adjoint velocity and adjoint pressure to have a well-posed system. Finally, from the integral over the boundary term in Eq. (19) it is clear one needs to evaluate the following expression

$$\frac{\partial J}{\partial \alpha} = \frac{\partial J}{\partial U_{inlet}} = \boldsymbol{n}(\overline{U}_{inlet} \cdot U_{inlet}) + \overline{U}_{inlet}(U_{inlet} \cdot \boldsymbol{n}) + 2\boldsymbol{v}_{eff}\boldsymbol{n} \cdot \mathbf{D}(\overline{U}_{inlet})$$
(32)

5 to compute the sensitivity.

**3.2.2 Wind Direction Effect**

As it was mentioned before, in ABL CFD simulations it is common to simulate first a 1D domain with a periodic boundary to obtain the inflow boundary condition. Then the cell center velocity of the 1D run is copied directly to its counterpart boundary face in the 3D domain. As a requirement, the number of cells in the 1D mesh and faces in the vertical direction of the 3D inflow boundary should be the same (see Figure 1). Moreover, and ideally, the face center heights in the 3D mesh are equal to their

10 boundary should be the same (see Figure 1). Moreover, and ideally, the face center heights in the 3D mesh are equal to their counterpart cell height in 1D. Although, in current work, a circular n inflow-outflow boundary is considered, with some small modification in the code the method can also be applied to other boundary shapes.

**Figure 1.** The inflow velocities of each cell from 1D precursor run (left) are copied to the boundary of the 3D domain (right) which has the similar number of cells in vertical direction. Ideally the height of each face in 3D domain boundary is exactly the same as its counterpart cell in the 1D mesh.

The inflow wind direction (WD) effect can be expressed by a rotation angle,  $\theta$ , which rotates the inflow from its default west to east (WD=270°) direction,

[revised manuscript text omitted]

---

## Referee Report (RR1)

**Review of "Adjoint-based Calibration of Inlet Boundary Condition for Atmospheric CFD Solvers"**
**Authors: Siamak Akbarzadeh, Hassan Kassem, Renko Buhr, Gerald Steinfeld, and Bernhard Stoevesandt**

The manuscript employs gradient- and adjoint-based method for calibration of inlet velocity profiles for ABL simulation. To that end, cost function for the optimization problem is defined such that the simulated velocity and the target velocity at the observation point in the simulation domain show maximum agreement for the optimized inflow profile. As a test case, the authors verify their method by applying it to two cases.

Accurate prediction and estimation of inflow condition is of great interest for wind farm simulation as well as for ABL research in general. Like many other methods (e.g. precursor simulation with re-scaled velocity field), gradient-based optimization of inflow profile can also be one of the approaches. Also if this method works perfectly, it will have advantage over other methods in terms of accuracy, as it does not make any big assumption.

The authors have addressed several issues raised by this and other reviews. They have also organized the manuscript, and it has become comparatively easy to follow. However, the manuscript need further modification. Please see the comments below.

**Specific comments:**

1. The abstract does not sufficiently describe the contents and findings of this manuscript. It is a very general description of what a reader can expect.

2. Pg. 2, line 25: it still mainly $\rightarrow$ it is still mainly

3. Pg. 6, line 1: $q \rightarrow \overline{p}$

4. Eq. (21): The first term $(-\nabla \overline{U} \cdot U)$ may not be correct. Please reconfirm it. In literature I could only find $-\left(\nabla \overline{U}\right)^{T} \cdot U$ and one another form of the cross-convective term.

5. Figure 10: Can you please also add the evolution of gradient as a function of adjoint iteration?

6. Conclusions: You should conclude the results and findings from your manuscript. Your current conclusions are too general.

---

## Author Response (AR2)

**The Authors response to the reviewers' comments**

The authors would like to thank the referees for their new comments. In the following, the comments will be addressed:

- **Anonymous Referee #2**

    1. Second part of (7): Is the LHS missing a transpose?

        Yes, a transpose sign was missing, but for the RHS term. Corrected.

    2. Equations (9) and (10): $R_1 - R_4$ are scalars. Remove boldface italic. Also, mention here that they are the components of $\boldsymbol{R}$ in (2)

        Modified accordingly.

    3. Equation (9): It is $U \cdot grad(U)$ , not $U \cdot grad \cdot U$

        Corrected and written as $(\boldsymbol{U} \cdot \nabla)\boldsymbol{U}$.

    4. Equation (9): $p$ is really a modified pressure.

        The word "modified" was added to the explanation of the variable $p$.

    5. Line 25: Do you mean $\bar{p}$, not $q$?

        Corrected. The adjoint variables are represented by variables with overbar line.

- **Anonymous Referee #4**

    1. The abstract does not sufficiently describe the contents and findings of this manuscript. It is a very general description of what a reader can expect.

        More information is added to both the abstract and the conclusion to better represent the findings of the study.

    2. Pg. 2, line 25: it still mainly $\rightarrow$ it is still mainly

        Corrected.

    3. Pg. 6, line 1: $q \rightarrow p$

        Corrected.

    4. Eq. (21): The first term $(-\nabla \bar{U} \cdot U)$ may not be correct. Please reconfirm it. In literature I could only find $-\nabla \bar{U}^T \cdot U$ and one another form of the cross-convective term.

        The term $-\nabla \bar{U} \cdot U$ in the manuscript is correct and similar to what has been derived in the original study of Othmer. Moreover, as mentioned now in the manuscript, a more detailed derivation of the adjoint system of equation for topology optimisation, which includes this term, can be found in the following study:
        Hinterberger, C. and Olesen, M.: Industrial application of continuous adjoint flow solvers for the optimization of automotive exhaust systems, CFD & Optimization, Antalya, Turkey, 2011.

Please also be aware that,

$$\boldsymbol{R} = (R_1, R_2, R_3, R_4)^T$$

$$\frac{dJ}{d\alpha} = \frac{\partial J}{\partial \alpha} + \overline{\boldsymbol{\psi}}^T \frac{\partial \boldsymbol{R}}{\partial \alpha}$$

$$\overline{\boldsymbol{\psi}}^T = (\overline{\boldsymbol{U}}, \overline{p})$$

5. Figure 10: Can you please also add the evolution of gradient as a function of adjoint iteration?

The plot of gradient history is now added to the Figure 10.

6. Conclusions: You should conclude the results and findings from your manuscript. Your current conclusions are too general.

More information is added to both the abstract and the conclusion to better represent the findings of the study.